# PERRY: Policy Evaluation with Confidence Intervals using Auxiliary Data

**Aishwarya Mandyam***  
*Stanford University*  
*am2@stanford.edu*

**Jason Meng***  
*Stanford University*  
*jiemeng@stanford.edu*

**Ge Gao**  
*Stanford University*  
*gegao@stanford.edu*

**Jiankai Sun**  
*Stanford University*  
*jksun@stanford.edu*

**Mac Schwager**  
*Stanford University*  
*schwager@stanford.edu*

**Barbara E. Engelhardt**  
*Gladstone Institutes*  
*Stanford University*  
*barbarae@stanford.edu*

**Emma Brunskill**  
*Stanford University*  
*ebrun@cs.stanford.edu*

**Reviewed on OpenReview:** *https://openreview.net/forum?id=7592*

## Abstract

Off-policy evaluation (OPE) methods estimate the value of a new reinforcement learning (RL) policy prior to deployment. Recent advances have shown that leveraging auxiliary datasets, such as those synthesized by generative models, can improve the accuracy of OPE methods. Unfortunately, such auxiliary datasets may also be biased, and existing methods for using data augmentation within OPE lack principled uncertainty quantification. In high stakes domains like healthcare, reliable uncertainty estimates are important for ensuring safe and informed deployment of RL policies. In this work, we propose two methods to construct valid confidence intervals for OPE with data augmentation. The first provides a confidence interval over $V^\pi(s)$, the policy value conditioned on an initial state $s$. To do so we introduce a new conformal prediction method suitable for Markov Decision Processes (MDPs) with continuous state spaces, extending prior work to higher-dimensional settings. Second, we consider the more common task of estimating the average policy performance over all initial states, $V^\pi$; we introduce a method that draws on ideas from doubly robust estimation and prediction powered inference. Across simulators spanning inventory management, robotics, healthcare, and a real healthcare dataset from MIMIC-IV, we find that our methods can effectively leverage auxiliary data and consistently produce confidence intervals that cover the ground truth policy values, unlike previously proposed methods. Our work enables a future in which OPE can provide rigorous uncertainty estimates for high-stakes domains.

# 1 Introduction

Off-policy evaluation (OPE) (Precup et al., 2000; Sutton & Barto, 2018) is used to estimate the value of a new (target) reinforcement learning (RL) policy prior to deployment using a historical (behavior) dataset from a distinct behavior policy. OPE is particularly important in high-stakes domains (Gottesman et al., 2020; Mandel et al., 2014; Fu et al., 2020), where directly deploying new policies without prior evaluation can be costly or even harmful to participants.

However, standard OPE methods frequently struggle when the target policy is very different from the behavior policy. This is due to limited dataset coverage (Jiang & Li, 2016). To address this, several recent works have proposed using synthetic auxiliary data to improve the coverage of the behavior dataset and subsequently the accuracy of OPE methods (Tang & Wiens, 2023; Gao et al., 2024; Mandyam et al., 2024). However, such approaches have either focused on the contextual bandit setting, or focused on promising empirical success in sequential settings; they lack formal assurances on the quality of the proposed estimates.

In high stakes, multi-step domains, it is often important to have confidence intervals (CIs) over the proposed policy value estimates. Such intervals support safer, more informed policy selection and deployment. Therefore, we argue that principled uncertainty quantification is necessary for OPE in RL in the emerging regime where both real and synthetic trajectories are used. While there is a notable body of prior work that developed CIs using *only* offline (real) data for OPE in RL (Thomas et al., 2015a;b; Taufiq et al., 2022; Foffano et al., 2023), to our knowledge, none provides guarantees in settings that combine offline and synthetic trajectories. In this paper we take steps towards addressing this gap.

We formalize uncertainty quantification for OPE with mixed (real and synthetic) behavior data and identify two settings that require uncertainty-aware OPE. First, in domains like healthcare, it is common for stakeholders to deliberate between decision-making policies to use for individuals that start in the same state: for example, a clinician may use the same treatment policy on all patients in the same stage of a disease. Estimating CIs for initial-state-conditioned policy performance is thus an important task that can benefit substantially from data augmentation. Our first method, `CP-Gen`, provides conformal prediction intervals for such state-conditioned values.

Second, we address evaluation of the target policy's expected value averaged over the distribution of initial states. This is relevant in settings where a single policy may be selected for the whole population, and a stakeholder wants to choose among different policies. We introduce a second method `DR-PPI`, which leverages techniques from doubly robust estimation and prediction-powered inference (Angelopoulos et al., 2023) to correct biases from synthetically generated trajectories and produce valid CIs.

Our empirical studies across inventory control, sepsis treatment, robotic control, and the MIMIC-IV electronic health record (EHR) dataset demonstrate that our methods, which can leverage synthetic data, can match or improve over state-of-the-art baselines that provide correct CIs using only real data. Our contributions follow.

1. **We formalize the problem of uncertainty quantification** for OPE in MDPs that leverage synthetically generated trajectories and introduce `CP-Gen` and `DR-PPI` (Section 3) for two natural settings where CIs are important.

2. **We prove that both methods yield valid CIs** and achieve the desired coverage probability either asymptotically or within a small margin of error for finite sample sizes (Section 4).

3. **We empirically evaluate the estimators** in four domains including a real-world EHR dataset, showing that our estimates, which leverage auxiliary synthetic data, produce CIs with the correct coverage that match or are tighter than baselines that do not use the auxiliary data. (Section 5).

## 2 Background

### 2.1 Problem setting

We consider a decision-making setting defined by the MDP $\mathcal{M} = (\mathcal{S}, \mathcal{A}, P, R, d_0, \gamma, H)$. $\mathcal{S}, \mathcal{A}$ denote the possibly infinite state and action spaces respectively. $P : \mathcal{S} \times \mathcal{A} \to \Delta(\mathcal{S})$ represents the transition dynamics, $R : \mathcal{S} \times \mathcal{A} \to \Delta(\mathbb{R})$ is the reward function, and $d_0 \in \Delta(\mathcal{S})$ is the initial state distribution. $\gamma$ is the discount factor and $H$ is the fixed horizon. A trajectory $\tau$ is defined as $\tau : \{s_t, a_t, r_t\}_{t=0}^H$ where $s_t, a_t, r_t$ are the state, action, and instantaneous reward observed at timestep $t$. The return of a trajectory $\tau$ is $J(\tau) = \sum_{t=1}^H \gamma^{t-1} r_t$ where $\tau \sim \pi$ and $\pi$ is the policy used to generate the trajectory. The value of the policy $V^\pi = \mathbb{E}_{\tau \sim \pi}[J(\tau)]$ is calculated as an expectation over the possible trajectories that could arise from $\pi$. The value of a policy conditioned on an initial starting state $s$ is $V^\pi(s) = \mathbb{E}_{\tau \sim \pi}[J(\tau)|s_0 = s]$.

### 2.2 Off-policy evaluation (OPE)

Our work focuses on the task of OPE, where the goal is to estimate the value of a target policy $\pi_e$ given a dataset of behavior trajectories $D_{\pi_b}$ that arise from a distinct behavior policy $\pi_b$. In a typical OPE setup, we assume access to $\pi_e$. In our work, we also assume $\pi_b$ is known similar to prior work (Thomas & Brunskill, 2016; Farajtabar et al., 2018), though we apply our methods empirically in settings where $\pi_b$ must be estimated.

There are several standard approaches for OPE, including importance sampling (IS) (Precup et al., 2000), direct method (DM) (Li et al., 2010; Beygelzimer & Langford, 2009; van Seijen et al., 2009; Harutyunyan et al., 2016; Le et al., 2019; Voloshin et al., 2021), and doubly robust (DR) approaches (Farajtabar et al., 2018; Dudík et al., 2011; Jiang & Li, 2016). IS-based estimators re-weigh each trajectory in the $D_{\pi_b}$ using an inverse propensity score (IPS) $\rho(\tau) = \prod_{t=1}^H \frac{\pi_e(a_t|s_t)}{\pi_b(a_t|s_t)}$. DM estimators learn a reward model using the behavior trajectories to directly estimate the value of the target policy. DR methods combine the advantages of IS and DM estimators and provide favorable guarantees when either the IPS ratio or the reward model is inaccurate.

### 2.3 Conformal Prediction

One strategy we consider for uncertainty quantification is conformal prediction, which is a framework for constructing prediction intervals with finite-sample coverage guarantees under minimal assumptions (Vovk et al., 2005). Given a dataset of i.i.d. samples $\{Z_i\}_{i=1}^n$ and a nonconformity score function $V(Z)$, CP constructs a prediction set $\mathcal{C}_{n,\alpha}$ such that

$$\mathbb{P}(Z_{n+1} \in \mathcal{C}_{n,\alpha}) \geq 1 - \alpha,$$

without requiring distributional assumptions beyond exchangeability.

In its simplest form, split conformal prediction partitions the data into a training set and a calibration set. A model is fit on the training set, and nonconformity scores $\{V_i\}_{i=1}^n$ are computed on the calibration set. The prediction interval is then constructed using empirical quantiles:

$$\mathcal{C}_{n,\alpha} = \{z : V(z) \leq Q_{1-\alpha}(\{V_i\}_{i=1}^n)\}.$$

In settings with distribution shift, such as OPE, the calibration data are not drawn from the target distribution. Weighted conformal prediction addresses this by assigning importance weights $w_i$ to each calibration sample (Tibshirani et al., 2019; Taufiq et al., 2022). The empirical distribution is replaced by a weighted empirical distribution

$$F_n(v) = \sum_{i=1}^n p_i \mathbf{1}\{V_i \leq v\}, \quad p_i = \frac{w_i}{\sum_{j=1}^n w_j + w_{n+1}},$$

where $w_{n+1}$ corresponds to the test point.

The resulting prediction set uses weighted quantiles

$$\mathcal{C}_{n,\alpha} = \{z : Q_{\alpha/2}(F_n) \leq V(z) \leq Q_{1-\alpha/2}(F_n)\}.$$

## 2.4 Prediction-Powered Inference

A related line of work addresses the problem of drawing valid statistical conclusions when labeled data is scarce but model predictions can be used to add dataset coverage. Prediction-powered inference (PPI) (Angelopoulos et al., 2023) is a framework for constructing valid confidence intervals by combining a small labeled dataset with a larger unlabeled dataset augmented with model predictions.

Let $\{(X_i, Y_i)\}_{i=1}^n$ denote a labeled dataset, where $X_i \in \mathcal{X}$ are inputs and $Y_i \in \mathbb{R}$ are observed outcomes. In addition, suppose we have access to a larger unlabeled dataset $\{X_j\}_{j=1}^N$, for which the corresponding outcomes are not observed. Let $\hat{f}(X)$ denote a prediction model trained to estimate $Y$ from $X$.

The goal is to estimate a population parameter of the form

$$\theta = \mathbb{E}[Y],$$

or more generally, $\theta = \mathbb{E}[g(X, Y)]$ for some function $g$. A prediction-powered estimator takes the form

$$\hat{\theta}_{\mathrm{PPI}} = \frac{1}{N} \sum_{j=1}^N \hat{f}(X_j) + \frac{1}{n} \sum_{i=1}^n \left( Y_i - \hat{f}(X_i) \right),$$

where the first term uses model predictions over the large unlabeled dataset for efficiency, and the second term corrects for bias using the labeled data. The PPI estimator can be interpreted as a bias-corrected plug-in estimator: the model-based term provides a low-variance estimate, while the residual correction ensures consistency even if the model $\hat{f}$ is misspecified. Under mild conditions, $\hat{\theta}_{\mathrm{PPI}}$ is asymptotically normal and admits valid confidence intervals.

In our work, we take inspiration from the construction of the standard PPI estimator, which yields an asymptotic CI. However, the problem setup in PPI is distinct from ours. PPI assumes that we have access to a large dataset of observations that are unlabeled; the role of the ML model is to label the observations. In contrast, in our setting, we must both generate synthetic samples (i.e., trajectories) and their corresponding labels (i.e., returns); this setting necessitates a distinct methodology.

## 2.5 Related Literature

**OPE with data augmentation**. As discussed in Section 1, standard OPE methods suffer when the behavior dataset has limited coverage. Because OPE methods are typically used with finite sample sizes, OPE estimates can be biased or have high variance (Precup et al., 2000; Jiang & Li, 2016; Thomas & Brunskill, 2016). To address this concern, several works have proposed using auxiliary information to enhance OPE estimators, using data augmentation either from a secondary dataset (Tang & Wiens, 2023; Mandyam et al., 2024) or by generating synthetic trajectories based on historical data (Gao et al., 2024; Sun et al., 2023; Gao et al., 2023).

Within model-based approaches, different classes of generative models have been considered. Some works learn transition models (e.g., Neural Networks (Chua et al., 2018) or Variational Autoencoders (VAEs) (Gao et al., 2024)) that enable step-wise rollout under a policy, while others employ trajectory-level generative models such as diffusion-based models (Sun et al., 2023) to generate entire trajectories. These approaches differ in how they model dynamics and generate samples, but share the common goal of improving coverage of the state-action space. These works find that leveraging auxiliary data can improve OPE estimates in some domains such as robotic control. However, these methods may introduce additional bias due to errors in the auxiliary data, and lack theoretical guarantees or rigorous uncertainty quantification for the MDP setting. In contrast, our work emphasizes uncertainty quantification, which can provide more information to effectively compare between policies.

**Conformal prediction for OPE**. There are several strategies to perform uncertainty quantification, including conformal prediction. Taufiq et al. (2022) first applied weighted conformal prediction to the OPE setting for contextual bandits. Foffano et al. (2023) later extended this work to create conformal intervals for OPE in the multi-step setting. Crucially, their approximation relies on an integral that is difficult to

compute and implement in setting with continuous state spaces. One of our proposed methods, `CP-Gen`, is inspired by the last approach, but uses a novel technique to compute the weights needed for conformal prediction, allowing us to use OPE in settings with continuous and higher-dimensional state spaces than prior work. In addition, prior work did not consider data augmentation. Our new approach achieves tighter CIs through careful use of auxiliary synthetic datasets.

## 3 Methods

In this work, we study a setting in which we have access to a real, offline dataset, and synthetically generated trajectories. In general, trajectories produced by generative models may be biased or drawn from a distribution distinct from $\pi_b$, which can introduce error and/or variance into the resulting OPE estimate for sequential decision processes. We propose two new methods for computing CIs for OPE in RL for two common settings where CIs would be beneficial. For clarity, we summarize key notation in a table that can be referenced throughout the paper (Table 1).

| Notation | Description |
|---|---|
| $J(\tau)$ | Return of the trajectory $\tau$, $\sum_{t=1}^{H} \gamma^{t-1} r_t$ where $\gamma$ is a discount factor |
| $V^\pi$ | Value of the policy $\pi$, $\mathbb{E}_{\tau \sim \pi}[J(\tau)]$, which is calculated as an expectation over trajectories sampled from the policy. |
| $V^\pi(s)$ | Value of the policy $\pi$ conditioned on a starting state $s$, $\mathbb{E}_{\tau \sim \pi}[J(\tau)|s_0 = s]$ |
| $p^{\pi_e}(\tau)$ | Probability of observing trajectory $\tau$ under the target policy $\pi_e$. |
| $\tilde{p}^{\pi_e}(\tau)$ | Probability of observing trajectory $\tau$ under the dynamics distribution of the generative model. |
| $\Delta_{rr'}$ | Return difference of a pair of trajectories, one from the original behavior dataset and one generated. For example, $J(\tau_i) - J(\tilde{\tau}_j)$ where $\tau_i$ is an observed trajectory and $\tilde{\tau}_j$ is a generated trajectory. $\Delta_{rr'}$ represents the random variable of return differences, and $\delta_{rr'}$ is the return difference for a single sample. |
| $\mathbb{P}^\pi_{(S,\Delta_{rr'})}$ | Joint distribution of the initial state $S$ and return-difference random variable $\Delta_{rr'}$ induced by trajectories drawn under the policy $\pi$. |
| $w(s, \delta_{rr'})$ | Weight associated with sample that has an initial-state $s$ and score (or return difference) $\delta_{rr'}$. |
| $w_\epsilon(s, \delta_{rr'})$ | Approximated weight for sample with initial-state $s$ and score $\delta_{rr'}$. |
| $\epsilon_s, \epsilon_r$ | Radius of ball around a given state $s$ and a given score $\delta_{rr'}$. |
| $F_n^{(s,\delta_{rr'})}$ | Weighted empirical cumulative distribution function (CDF) of calibration scores (return differences) evaluated at $(s, \delta_{rr'})$, constructed using normalized importance weights. |
| $\text{score}_i^{(s,\delta_{rr'})}$ | Non-conformity score for sample $i$, equal to the return difference of the paired trajectories, i.e. $\text{score}_i = \Delta_{rr',i} = J(\tau_i) - J(\tilde{\tau}_i)$. |
| $p_i^w(s, \delta_{rr'})$ | Normalized weight assigned to sample $i$ in the weighted conformal procedure; proportional to $w(S_i, \Delta_{rr',i})$ and normalized so weights sum to one (including the test point). |
| $C_{n,\alpha}(s)$ | Weighted conformal prediction band for the return difference at state $s$, defined by the central $(1 - \alpha)$ weighted quantiles of $F_n^{(s,\delta_{rr'})}$. |
| $\hat{C}_{n,\alpha}(s)$ | Estimated conformal interval conditioned on an initial-state $s$ learned using $n$ offline trajectories with $1 - \alpha$ confidence. |
| $\widehat{C}_\alpha$ | Estimated confidence interval with confidence level $(1 - \alpha)$. |
| $d_0$ | Initial state-distribution |
| $\tilde{J}(\tau)$ | Re-weighted return of trajectory $\tau \sim \pi_e$. Specifically, $\tilde{J}(\tau) = \rho(\tau) * J(\tau)$. |
| $D_1, D_2$ | Two splits of the behavior dataset $D_{\pi_b}$ |

Table 1: Reference table for notation used throughout the paper.

### 3.1  `CP-Gen`: Confidence Intervals for OPE from a Starting State

First, we consider a setting in which our goal is to estimate an initial-state-conditioned policy value $V^{\pi_e}(s)$. Estimating state-conditioned policy values has been understudied in the OPE literature, which tends to focus on estimators that average performance over the full population of initial states. The task of estimating state-conditioned policy values can benefit from data augmentation, since data from individual starting states is sparse. To address this, we propose `CP-Gen`, a new conformal prediction method for OPE.

Given an initial state $s$, we estimate $V^{\pi_e}(s)$ by carefully manipulating the definition of the policy value as follows.

$$V^{\pi_e}(s) = \mathbb{E}_{\tau \sim p^{\pi_e}|s_0=s}\left[J(\tau)\right] \tag{1}$$

$$= \sum_{\tau \sim p^{\pi_e}|s_0=s} p^{\pi_e}(\tau)J(\tau) \tag{2}$$

$$= \underbrace{\sum_{\tilde{\tau} \sim \tilde{p}^{\pi_e}|s_0=s} \tilde{p}^{\pi_e}(\tilde{\tau})J(\tilde{\tau})}_{\text{simulator estimate}} + \underbrace{\left[\sum_{\tau \sim p^{\pi_e}|s_0=s} p^{\pi_e}(\tau)J(\tau) - \sum_{\tilde{\tau} \sim \tilde{p}^{\pi_e}|s_0=s} \tilde{p}^{\pi_e}(\tilde{\tau})J(\tilde{\tau})\right]}_{\text{model bias / return discrepancy}} \tag{3}$$

Here, $\tau \sim p^{\pi_e}$ is a trajectory drawn from the dynamics distribution associated with $\pi_e$ and $p^{\pi_e}(\tau)$ is the probability of observing trajectory $\tau$ under the policy $\pi_e$. In Equation (3), we add and subtract the simulator estimate of the state-conditioned policy value $\sum_{\tilde{\tau} \sim \tilde{p}^{\pi_e}|s_0=s} \tilde{p}^{\pi_e}(\tilde{\tau})J(\tilde{\tau})$, where $\tilde{p}$ is the dynamics distribution induced by the generative model and $\tilde{\tau}$ is a synthetic trajectory sampled from a generative model that approximates the dynamics distribution of the target policy, $\tilde{p}^{\pi_e}$.

We can now approximate the "model bias/return discrepancy" term using empirical averages. We note that the empirical averages require access to trajectories that are sampled from the target policy distribution, $p^{\pi_e}$. The "model bias/return discrepancy" term can be approximated as

$$\approx \sum_{\tilde{\tau} \sim \tilde{p}^{\pi_e}|s_0=s} \tilde{p}^{\pi_e}(\tilde{\tau})J(\tilde{\tau}) + \underbrace{\frac{1}{n}\sum_{i=1}^{n} J(\tau_i|s_0=s) - \frac{1}{nM}\sum_{j=1}^{nM} J(\tilde{\tau}_j|s_0=s)}_{\text{approximate the expected value by empirical average}} \tag{4}$$

$$= \sum_{\tilde{\tau} \sim \tilde{p}^{\pi_e}|s_0=s} \tilde{p}^{\pi_e}(\tilde{\tau})J(\tilde{\tau}) + \frac{1}{nM}\sum_{i=1}^{n}\sum_{j=1}^{M} \underbrace{(J(\tau_i|s_0=s) - J(\tilde{\tau}_{ij}|s_0=s))}_{\text{return difference of a pair of trajectories}}, \tag{5}$$

where $n/M$ is the number of behavior/synthetic trajectories, and $J(\tau|s_0=s)$ is the return of trajectory $\tau$ given initial state $s$.

Inspired by conformal prediction for regression, our goal is to produce an interval $\hat{C}_{n,\alpha}(s)$, which is specific to initial state $s$ and the number of offline behavior trajectories $n$. This interval defines a band such that, with high probability, the return difference between any offline trajectory and its corresponding generated trajectory that starts from the same initial state $s$ ("return difference of a pair of trajectories" in Equation (5)) lies within the band. More specifically,

$$P^{\pi_e}\left(\underbrace{J(\tau|s_0=s) - J(\tilde{\tau}|s_0=s)}_{\text{return difference of a pair of trajectories}} \in \hat{C}_{n,\alpha}(s)\right) \geq 1 - \alpha, \tag{6}$$

where $P^{\pi_e}$ is the probability measure induced by the target policy $\pi_e$ and $\alpha$ is the confidence level. Given this goal, the final conformal prediction interval for the value of the initial state $s$, $V^{\pi_e}(s)$, is

$$\sum_{\tilde{\tau} \sim \tilde{p}^{\pi_e}, s_0=s} \tilde{p}^{\pi_e}(\tilde{\tau})J(\tilde{\tau}) + \hat{C}_{n,\alpha}(s). \tag{7}$$

*Remark* (Conformal band intuition). *The role of conformal prediction in* `CP-Gen` *is to construct a distribution-free confidence interval for the return discrepancy between real and synthetic trajectories. By calibrating the return differences $J(\tau) - J(\tilde{\tau})$ using both offline and generated data, and reweighting to correct for the policy distribution shift between $\pi_b$ and $\pi_e$, we obtain an interval that provably covers the unknown bias term. Adding this interval to the synthetic estimate therefore yields a valid confidence interval for $V^{\pi_e}(s)$.*

---

**Algorithm 1** `CP-Gen`

---

**Require:** Offline dataset $\mathcal{D}_{\pi_b}$, behavior policy $\pi_b$, target policy $\pi_e$, initial state $x$.

1: Split $D_{\pi_b}$ (size $n$) into $D_{tr}$ ($n/2$) and $D_{cal}$ ($n/2$)
2: Fit a generative model $\mathcal{T}$ using $D_{tr}$.
3: For each trajectory $\tau_i \in D_{tr}$, generate $M$ trajectories $\{\tilde{\tau}_{i,m}\}_{m=1}^M$ under $\pi_b$ with the same initial state as $\tau_i$, record the pairs as $\{(\tau_i, \tilde{\tau}_{i,m})\}_{m=1}^M$.
4: For each trajectory $\tau_j \in D_{cal}$, generate $N$ trajectories $\{\tilde{\tau}_{j,k}\}_{k=1}^N$ under $\pi_b$ with the same initial state as $\tau_j$, record the pairs as $\{(\tau_j, \tilde{\tau}_{j,k})\}_{k=1}^N$.
5: For each $(\tau_j, \tilde{\tau}_{j,k})$, calculate the weight $\hat{w}_\epsilon(x_j, J(\tau_j) - J(\tilde{\tau}_{j,k}))$ using $(\tau_i, \tilde{\tau}_{i,m})$ (Equation (10)).
6: Given an initial state $x$, calculate $p_{j,k}^{\hat{w}}(x,y)$ and $p_{\frac{nN}{2}+1}^{\hat{w}}$ using Equation (13).
7: For each $(\tau_j, \tilde{\tau}_{j,k})$, calculate the score $\text{score}_{j,k} = J(\tau_j) - J(\tilde{\tau}_{j,k})$.
8: Calculate $F^{(x,y)}$ using Equation (12).
9: Calculate confidence interval $\hat{C}_{n,\alpha}$ over the value of trajectories starting in initial state $x$ using Equation (11).
10: Rollout trajectories under $\pi_e$ from $\mathcal{T}$ and get the first term in Equation (7).

---

The derivation above assumes access to trajectory pairs $(\tau, \tilde{\tau})$ drawn under the target policy $\pi_e$. However, in practice, we only observe real trajectories from the behavior policy $\pi_b$. To construct comparable pairs, we generate synthetic trajectories $\tilde{\tau}$ using the learned model while conditioning on initial states observed under $\pi_b$, and rolling out using $\pi_b$. As a result, the empirical return differences $J(\tau \mid s_0 = s) - J(\tilde{\tau} \mid s_0 = s)$ are drawn from a distribution induced by $\pi_b$, rather than $\pi_e$. To bridge this gap, we apply weighted conformal prediction, which reweights these samples to approximate the distribution of return differences under $\pi_e$, enabling valid inference despite the distribution shift.

Now, we use conformal prediction to calculate the band. Unlike standard conformal prediction, we must address the distribution shift induced by the difference between the behavior and target policies. To do so, prior work (Foffano et al., 2023), which builds on related work (Tibshirani et al., 2019; Taufiq et al., 2022), proposed CP methods for MDPs that weigh the calibration scores using estimates of the likelihood ratio.

However, this prior work does not consider the use of generated trajectories. Therefore we introduce a new sample reweighting technique that accounts for the distribution shift (between $\pi_b$ and $\pi_e$) in both the real and generated trajectories (see full derivation in Appendix E). To simplify notation, let $s \in S$ be the initial state and $\delta_{rr'} \in \Delta_{rr'}$ be the return difference of a pair of trajectories (one from the original behavior dataset, and one generated). Then, the weight for a given sample is

$$w(s, \delta_{rr'}) := \mathbb{P}_{(S, \Delta_{rr'})}^{\pi_e}(s, \delta_{rr'}) / \mathbb{P}_{(S, \Delta_{rr'})}^{\pi_b}(s, \delta_{rr'}) \tag{8}$$

$$= \mathbb{E}_{\tau \sim p^{\pi_b}, \tilde{\tau} \sim \tilde{p}^{\pi_b}} \left[ \underbrace{\frac{\prod_{t=1}^H \pi_e(a_t|s_t)\pi_e(\tilde{a}_t|\tilde{s}_t)}{\prod_{t=1}^H \pi_b(a_t|s_t)\pi_b(\tilde{a}_t|\tilde{s}_t)}}_{\substack{\text{product of IPS ratios}\\\text{of real and generated trajectories}}} \mid \underbrace{s_0 = s, \delta_{J(\tau)J(\tilde{\tau})} = \delta_{rr'}}_{\substack{\text{conditioned on same initial state}\\\text{and reward difference}}} \right]. \tag{9}$$

This weight is an expectation of the IPS ratio over all observations that share the same input ($s$) and score ($\delta_{rr'}$). However, calculating $w(s, \delta_{rr'})$ will become intractable as the size of the MDP increases, because we may not have access to many trajectories that share $s$ and $\delta_{rr'}$.

To mitigate this, and allow us to compute valid conformal prediction intervals in continuous state and action spaces, we use something we refer to as "$\epsilon-$approximation" to estimate the weight for a given sample. With

$\epsilon$-approximation, we can approximate the weight as

$$w_\epsilon(s, \delta_{rr'}) = \mathbb{E}_{\tau \sim p^{\pi_b}, \tilde{\tau} \sim \tilde{p}^{\pi_b}} \left[ \frac{\prod_{t=1}^H \pi_e(a_t|s_t)\pi_e(\tilde{a}_t|\tilde{s}_t)}{\prod_{t=1}^H \pi_b(a_t|s_t)\pi_b(\tilde{a}_t|\tilde{s}_t)} \Big| \underbrace{s_0 \in B(s, \epsilon_s), \delta_{J(\tau)J(\tilde{\tau})} \in B(\delta_{rr'}, \epsilon_r)}_{\epsilon\text{-balls around } s \text{ and } \delta_{rr'}} \right], \quad (10)$$

where $B(s, \epsilon_s)$ represents a ball around the input $s$ of radius $\epsilon_s$ and $B(\delta_{rr'}, \epsilon_r)$ is a ball around the output $\delta_{rr'}$ of radius $\epsilon_r$. This setup allows for small perturbations around $s$ and $\delta_{rr'}$ when aggregating samples. In particular, $B(s, \epsilon_s)$ captures any input $s$ that is within a small distance $\epsilon_s$ of $s$, and likewise for $B(\delta_{rr'}, \epsilon_r)$.

*Remark* (Weight approximation in continuous state-space MDPs). *The weight $w(s, \delta_{rr'})$ (Equation (8)) is a population quantity that correctly accounts for distribution shift between the behavior and evaluation policies for both observed and synthetically generated trajectories. If this weight were known exactly, `CP-Gen` would yield exact conformal prediction intervals. However, in continuous state and return spaces, it is likely that the event $\{S = s, \Delta_{rr'} = \delta_{rr'}\}$ has probability zero, making direct estimation of $w(s, \delta_{rr'})$ infeasible from finite data.*

*To address this, we introduce "$\epsilon$-approximation," which replaces exact conditioning with conditioning on local neighborhoods. Specifically, the balls $B(s, \epsilon_s)$ and $B(\delta_{rr'}, \epsilon_r)$ collect trajectories whose initial states and return differences are close to $(s, \delta_{rr'})$, thereby pooling nearby samples to enable stable estimation. This approximation trades a controlled amount of bias for statistical tractability, which enables the application of conformal prediction in MDPs with continuous or infinite state and action space. Our theoretical analysis in Section 4 shows that the resulting loss in coverage is explicitly bounded as a function of $\epsilon_s$ and $\epsilon_r$. We emphasize that this $\epsilon$-based reweighting is a key technical contribution of `CP-Gen`, enabling conformal OPE with synthetic trajectories in settings where exact likelihood-ratio weighting is computationally or statistically infeasible.*

Now, using the weights $w_\epsilon$, the conformal band is

$$\hat{C}_{n,\alpha}(s) = \{\delta_{rr'} : \underbrace{Q(\frac{\alpha}{2}, F_n^{(s,\delta_{rr'})})}_{\frac{\alpha}{2} \text{ quantile of the CDF } F_n} \leq \underbrace{\text{score}_{n+1}^{(s,\delta_{rr'})}}_{\text{score of this pair of trajectories}} \leq \underbrace{Q(1 - \frac{\alpha}{2}, F_n^{(s,\delta_{rr'})})}_{(1 - \frac{\alpha}{2}) \text{ quantile of the CDF } F_n} \}, \quad (11)$$

where

$$F_n^{(s,\delta_{rr'})}(v) = \sum_{i=1}^n \underbrace{p_i^w(s, \delta_{rr'})}_{\text{weighted quantile}} \mathbb{1}\{\text{score}_i \leq v\} + \underbrace{p_{n+1}^w(s, \delta_{rr'})}_{\text{weighted quantile}} \mathbb{1}\{\infty \leq v\}, \quad (12)$$

$$p_i^w(s, \delta_{rr'}) = \begin{cases} \frac{w(S_i, \Delta_{rr',i})}{\sum_{j=1}^n w(S_j, \Delta_{rr',j}) + w(s, \delta_{rr'})} & \text{if } i \leq n, \\ \frac{w(s, \delta_{rr'})}{\sum_{j=1}^n w(S_j, \Delta_{rr',j}) + w(s, \delta_{rr'})} & \text{if } i = n+1. \end{cases} \quad (13)$$

Additionally, $Q$ is a quantile function and $\text{score}_i^{(S_i, \Delta_{rr',i})} = \Delta_{rr',i}$. We also note that typical conformal prediction methods do not provide coverage guarantees for individual samples. In our setting, however, the target of interest is $V^\pi(s)$, which is itself an expectation, so marginal coverage is sufficient.

*Remark* (Construction of the conformal band). *The conformal band in Equation (11) is constructed by applying weighted conformal prediction to the return differences $\Delta_{rr'}$. The weight $w(s, \delta_{rr'})$ plays the role of a likelihood-ratio correction, ensuring that calibration scores computed under the $\pi_b$ are properly reweighted to reflect the distribution induced by $\pi_e$.*

*The weighted empirical distribution function $F_n^{(s,\delta_{rr'})}$ aggregates these calibrated scores using normalized weights $p_i^w$, and includes an additional mass at $+\infty$ corresponding to the test point, as in standard conformal prediction. The confidence band $\hat{C}_{n,\alpha}(s)$ is then defined by the central $(1 - \alpha)$ quantiles of this weighted distribution. By construction, this guarantees that a new return difference drawn under the target policy falls within the band with high probability.*

*Finally, adding the conformal band to the synthetic estimate in Equation (7) propagates this uncertainty to the state-conditioned policy value $V^{\pi_e}(s)$, yielding a valid confidence interval that accounts for both the distribution shift observed in OPE and model bias.*

---

**Algorithm 2** `DR-PPI`

---

**Require:** Offline dataset $\mathcal{D}_{\pi_b}$, behavior policy $\pi_b$, target policy $\pi_e$.
 1: Split $D_{\pi_b}$ (size $n$) into $D_1$ and $D_2$ (each with size $\frac{n}{2}$).
 2: Fit a generative model $f_1$ using $D_1$.
 3: Use $f_1$ to generate $N_f$ rollouts $\{\tilde{\tau}_i\}_{i=1}^{N_f}$ from $\pi_e$.
 4: For each $\tau_j \in D_2$, use $f_1$ to generate $M$ rollouts $\{\tilde{\tau}_{m,j}\}_{m=1}^{M}$ with the same initial state $s_{0,j}$.
 5: Estimate $\widehat{V}_{\text{DR-PPI:1}}$ using Equation (14).
 6: Fit a generative model using $D_2$, and estimate $\widehat{V}_{\text{DR-PPI:2}}$ in the same way.
 7: Estimate $\hat{V}^{\pi_e}$ using Equation (15).
 8: Estimate the variance of $\hat{V}^{\pi_e}$ using Equation (16).
 9: Provide confidence interval $\hat{C}_\alpha$ using Equation (17).

---

### 3.2 `DR-PPI`: Confidence Intervals for Unconditional OPE Value Estimation

In Section 3.1, we derived a valid confidence interval for the initial-state conditioned policy value. Now, we study the more common task in OPE, which is to estimate the policy value averaged over the initial state distribution. A natural approach would be to aggregate the `CP-Gen` estimates across initial states, for example by applying a union bound. Unfortunately, this approach would result in confidence intervals that are impractically wide. Thus, we introduce a second estimator tailored to this setting, `DR-PPI`, which builds on ideas from doubly robust estimation and prediction-powered inference. Our goal is to construct an estimator of $V^{\pi_e} = \mathbb{E}_{s_0}[V^{\pi_e}(s_0)]$ with valid confidence intervals.

First, we assume that the initial state distribution $d_0$ is known (though our results extend to settings in which $d_0$ must be estimated). Now, we construct a cross-fitted, doubly-robust estimate of the policy value $V^{\pi_e}$ as follows. First, we split the behavior dataset $D_{\pi_b}$ into two equal parts, which we refer to as $D_1$ and $D_2$. We first use $D_1$ to fit a generative model $f_1$; this procedure is agnostic to the generative model used, and reasonable approaches include a diffusion model or a variational auto-encoder (VAE). Then, we use $f_1$ to generate $N_f$ rollouts $\{\tilde{\tau}_i\}_{i=1}^{N_f}$ where each rollout $\tilde{\tau}_i$ contains actions sampled from the target policy $\pi_e$. The rollouts are then used to calculate the model-based return; however since we expect this return to be biased, we add a correction term using the trajectories observed in $D_2$ as follows:

$$\widehat{V}_{\text{DR-PPI:1}}^{\pi_e} = \underbrace{\frac{1}{N_f}\sum_{i=1}^{N_f} J(\tilde{\tau}_i)}_{\text{model-based term}} + \underbrace{\frac{1}{n/2}\sum_{j \in D_2}\left(\tilde{J}(\tau_j) - \frac{1}{M}\sum_{m=1}^{M} J(\tilde{\tau}_{m,j} \mid s_{0,j})\right)}_{\text{correction term}}. \tag{14}$$

The correction term also uses synthetic trajectories. For each behavior trajectory $\tau_j$, we generate $M$ synthetic trajectories $\{\tilde{\tau}_{m,j}\}_{m=1}^{M}$, where each $\tilde{\tau}_{m,j}$ starts from the same initial state $s_{0,j}$ as $\tau_j$ and is generated using $f_1$, with actions sampled from $\pi_e$.

In the `DR-PPI` estimator, $n$ is the number of original behavior trajectories, and $\tilde{J}(\tau_i)$ is the re-weighted return of the behavior trajectory $\tau_i$. There are several possible ways to perform this re-weighting: IS (e.g., $\tilde{J}(\tau_i) = \rho(\tau_i)J(\tau_i)$), weighted IS (WIS), and per-decision IS (PDIS). Because the trajectory $\tau_i$ arises from the behavior policy, the re-weighting technique allows us to estimate its value as if it was generated from the target policy. Each re-weighting technique involves different bias-variance tradeoffs that are well-studied in the literature, and the preferred choice will depend on the horizon length and dataset size of the specific application. Regardless of the re-weighting technique, our asymptotic theoretical results hold.

To ensure that the data is used efficiently, we use cross-fitting (Chernozhukov et al., 2018) with two splits of the data. $\widehat{V}_{\text{DR-PPI:1}}$ uses $D_1$ to fit the generative model $f_1$ and uses $D_2$ to provide the correction. Similarly, we fit the generative model on $D_2$ to produce $f_2$ and correct the estimator using $D_1$, which yields $\widehat{V}_{\text{DR-PPI:2}}$. The final estimate (Algorithm 2) is the average of $\widehat{V}_{\text{DR-PPI:1}}$ and $\widehat{V}_{\text{DR-PPI:2}}$. The variance of the estimator can then

be calculated by combining plug-in estimates of the variance of the model-based term and the correction term for each dataset split.

In particular, we average the outcomes of $\widehat{V}_{\texttt{DR-PPI:1}}$ and $\widehat{V}_{\texttt{DR-PPI:2}}$ as follows,

$$\widehat{V}_{\texttt{DR-PPI}} = (\widehat{V}_{\texttt{DR-PPI:1}} + \widehat{V}_{\texttt{DR-PPI:2}})/2. \tag{15}$$

The variance of the estimator can be calculated using plug-in estimates as follows,

$$\mathbb{V}\left[\widehat{V}_{\texttt{DR-PPI}}\right] = \frac{1}{4}\left(\frac{\widehat{\sigma}_{f_1}^2}{N_f} + \frac{\widehat{\sigma}_{b_1}^2}{n/2} + \frac{\widehat{\sigma}_{f_2}^2}{N_f} + \frac{\widehat{\sigma}_{b_2}^2}{n/2}\right), \tag{16}$$

where $\sigma_f^2 = \mathbb{V}_{\tilde{\tau} \sim \tilde{p}^{\pi_e}}\left[J(\tilde{\tau})\right]$, and $\sigma_b^2 = \mathbb{V}_{\tau \sim p^{\pi_b}, \tilde{\tau} \sim \tilde{p}^{\pi_e}}\left[\tilde{J}(\tau) - \frac{1}{M}\sum_{m=1}^{M} J(\tilde{\tau}_m | s_0(\tau))\right]$.

Using this variance, an approximate CI for a given choice of coverage $(1 - \alpha)$ is

$$\widehat{C}_\alpha = \widehat{V}_{\texttt{DR-PPI}}^{\pi_e} \pm z_{1-\alpha/2} \sqrt{\mathbb{V}\left[\widehat{V}_{\texttt{DR-PPI}}^{\pi_e}\right]} \tag{17}$$

where $\mathbb{V}\left[\widehat{V}_{\texttt{DR-PPI}}^{\pi_e}\right]$ is the variance of the OPE estimate learned by $\texttt{DR-PPI}$, and $z_{1-\alpha/2}$ is the z-score corresponding to the $1 - \alpha/2$ quantile of the standard normal distribution.

We observe that $\texttt{DR-PPI}$ differs from traditional PPI in two key ways. First, in PPI, one typically has access to large quantities of unlabeled data, and an ML model is used to predict labels for these samples. In contrast, in our setting, the ML model (e.g., a generative model) is used to produce new samples, which can then be comparatively easily labeled via a reward function. Second, our setting involves distribution shift; we observe trajectories generated by a behavior policy, while our goal is to infer the value of a target policy that induces a different trajectory distribution. In contrast, standard PPI assumes that labeled and unlabeled samples are drawn from the same underlying distribution.

Before discussing the practical considerations when fitting the estimators, we first compare their constructions. When all trajectories in an environment begin from the same initial state, the point estimates of both methods are identical, differing only in their confidence intervals. The re-weighting schemes, however, are distinct: $\texttt{DR-PPI}$ re-weights only the real behavior trajectories, whereas $\texttt{CP-Gen}$ uses a product of IPS ratios averaged across a set of trajectories. Finally, the return differences used to compute the CI in $\texttt{CP-Gen}$ may exhibit higher variance than subtracting the mean of a set of trajectories from the return of a single trajectory in $\texttt{DR-PPI}$. However, this effect depends on the stochasticity of the generated trajectories and may vary across domains.

### 3.3 Practical considerations

There are several practical considerations to enable OPE in environments with large state and action spaces as well as settings in which $\pi_b$ and $\pi_e$ differ substantially. First, it is occasionally necessary to clip the largest IPS ratios to avoid extremely large intervals. Ionides (2008) shows that using a clip constant set to $n^{1/2}$ where $n$ is the number of dataset samples, provides an optimal first order rate in the resulting mean-squared error of the OPE estimator, balancing the bias introduced by the clipping with the variance reduction. This clipping constant also ensures the resulting estimate is still consistent. Following this, we set the clipping constant at a rate of $n^{1/2}$. We note that clipping typically introduces an additional bias to the theoretical results, which we do not account for in this work and leave to future work.

Additionally, in Algorithm 2, we propose splitting the behavior dataset into two portions and aggregating the OPE estimate calculated using each portion. If a pre-trained generative model is available, we use the full dataset to construct the CI, and no data splitting for generative model training is necessary. However, if no pre-trained model exists, we divide the data and use one half train the generative model, and the other half to compute the CI. Because these two subsets are independent, this preserves the exchangeability criterion for conformal prediction and the validity of $\texttt{DR-PPI}$. However, in practice, it may not possible to split the

behavior dataset due to its size. For these settings, we choose not to perform cross-fitting, and instead report results without sub-dividing the dataset. As discussed in Section 5, this can still result in valid, but higher variance CIs.

Finally, for `CP-Gen`, we must set $\epsilon_s$ and $\epsilon_r$ depending on the environment. We view $\epsilon_s$ and $\epsilon_r$ as hyperparameters that need. One way to do this is via cross-validation, where we split the behavior dataset into training and validation sets, and choose the $\epsilon_r, \epsilon_s$ that yields the most accurate estimate of the value function $V^{\pi_b}(s)$ on the validation set.

## 4  Theoretical Results

Now, we discuss the theoretical guarantees of our approaches. As is standard in prior OPE literature, we assume that the target and behavior policies share common support, and that the instantaneous rewards and IPS ratios are bounded (Farajtabar et al., 2018; Thomas & Brunskill, 2016).

### 4.1  `CP-Gen` produces valid conformal prediction intervals

We make a few additional assumptions to analyze `CP-Gen`. These assumptions balance theoretical rigor with practical relevance, allowing us to derive meaningful guarantees settings with continuous state spaces. Importantly, they still encompass a broad class of real-world MDPs.

**Assumption 1** (Lipschitz Continuity of the Policy). There exist constants $L_\pi, L_{\pi,s}, L_{\pi,a}$ such that for $\pi \in \{\pi_b, \pi_e\}$ and all $s, s_1 \in \mathcal{S}, a, a_1 \in \mathcal{A}$,

$$TV(\pi(\cdot|s), \pi(\cdot|s_1)) \leq L_\pi \|s - s_1\| \tag{18}$$

$$|\pi(a|s) - \pi(a_1|s_1)| \leq L_{\pi,s}\|s - s_1\| + L_{\pi,a}\|a - a_1\|. \tag{19}$$

**Assumption 2** (Lipschitz Transition Dynamics). For all $s, s_1 \in \mathcal{S}, a, a_1 \in \mathcal{A}$,

$$TV(p(\cdot|s,a), p(\cdot|s_1,a_1)) \leq L_{p,s}\|s - s_1\| + L_{p,a}\|a - a_1\|. \tag{20}$$

**Assumption 3** (Score Smoothness). The map $(s, \delta_{rr'}) \mapsto w(s, \delta_{rr'})$ is $L_r$-Lipschitz in its second argument: $|w(s, \delta_{rr'}) - w(s, \delta'_{rr'})| \leq L_r|\delta_{rr'} - \delta'_{rr'}|$.

We consider Assumptions 1 and 2 mild. In most cases, Assumption 1 holds with a sufficiently large Lipschitz constant; in practice, these constants are small when similar states are assigned similar actions, a condition often justified in domains like healthcare, where similar patients receive similar treatments. A comparable assumption has been studied in prior work (Liu et al., 2022). Similarly, Assumption 2 is a smoothness assumption on the transition dynamics which has been used in prior work (Asadi et al., 2018). For example, in healthcare, patients with comparable clinical profiles often respond similarly to similar treatments; small changes in dosage or patient characteristics typically produce gradual, not abrupt, differences in outcomes.

Assumption 3 requires that the return differences between trajectories are smooth in their expected IPS ratios. However, in domains with a large number of samples, where we can use a more fine-grained $\epsilon_r$, the Lipschitz assumption here (which is multiplied by $\epsilon_r$ in our theoretical bound) will have much less impact. Overall, our assumptions are used to account for potential errors introduced by $\epsilon$-approximation, used in large or continuous state spaces and ensure that the resulting averaging error is bounded.

Under the stated assumptions, we now demonstrate that `CP-Gen` produces valid conformal prediction intervals within a small margin of error (Theorem 1, proof in Appendix E).

**Theorem 1** (`CP-Gen` calculates a valid conformal prediction interval). *Under Assumptions 1 to 3, suppose that $\mathbb{E}_{\pi_b}[|\hat{w}_\epsilon(S, \Delta_{rr'})|^k] \leq d^{2k}$ for some $k \geq 2$ and finite $d$. The conformal band has a lower bounded coverage*

$$P^{\pi_e}(\Delta_{rr'} \in \hat{C}_{n,\alpha}(S)) \geq 1 - \alpha - \Delta_w, \tag{21}$$

*where $\Delta_w = \frac{1}{2}\mathbb{E}^{\pi_b}|\hat{w}_\epsilon(S, \Delta_{rr'}) - w(S, \Delta_{rr'})|$ is the estimation error with scale*

$$\Delta_w = \tilde{\mathcal{O}}(n^{-1/2}\epsilon_s^{-3d_s/2}\epsilon_r^{-3/2} + \epsilon_s + \epsilon_r), \tag{22}$$

where $d_s$ is the dimension of $\mathcal{S}$.

In addition, if the non-conformity scores $\{V_i\}_{i=1}^n$ have no ties almost surely, then

$$P^{\pi_e}(\Delta_{rr'} \in \hat{C}_{n,\alpha}(S)) \leq 1 - \alpha - \Delta_w + cn^{1/k-1} \tag{23}$$

for some positive constant $c$ depending on $d$ and $k$ only.

*Remark* (Implications). *Theorem 1 shows that $\epsilon$-approximation results in a loss of coverage specified by $\Delta_w$, which depends primarily on $\epsilon_s$ and $\epsilon_r$. In environments where these constants are small, or there are a large number of samples, or $\epsilon_s, \epsilon_r$ are optimally selected, we can get a smaller loss of coverage. We also note that the guarantee is similar in form to prior conformal intervals for MDPs (Foffano et al., 2023), but our construction has significant benefits over prior work: it can leverage synthetic data and allows us to compute conformal bands for continuous states with our approximation of w.*

### 4.2  `DR-PPI` produces asymptotically valid confidnece intervals

In Section 3.2, we mention several choices for the importance-sampling correction including IS, WIS, and PDIS. Regardless of the correction style, we achieve asymptotically valid CIs (Theorem 2, proof in Appendix E).

**Theorem 2** (`DR-PPI` calculates an asymptotically valid CI)**.** *For all possible corrections $\tilde{R}_{IS}$, $\tilde{R}_{WIS}$ and $\tilde{R}_{PDIS}$,*

$$\liminf_{n,M,N_f \to \infty} P(V^{\pi_e} \in \hat{C}_\alpha) \geq 1 - \alpha. \tag{24}$$

*Remark* (Implications). Theorem 2 guarantees that `DR-PPI` produces confidence intervals with correct asymptotic coverage even when the generative model is misspecified. In particular, incorporating synthetic trajectories does not compromise validity, provided the importance-sampling correction is consistent. Different choices of correction affect efficiency in finite samples, but does not affect asymptotic coverage. Furthermore, the normal approximation underlying the confidence interval is well-suited to state value estimates, which are bounded and averaged across the state distribution; a Student-t correction could be adopted as a drop-in replacement for small samples if warranted by the application.

## 5  Empirical Results

Our theoretical analyses demonstrated that `CP-Gen` and `DR-PPI` can calculate valid CIs under mild assumptions. To complement this analysis, we seek to answer the following questions using empirical results: **1)** Does the $\epsilon$-approximation used in `CP-Gen` cause the estimated interval to be biased? **2)** Do `DR-PPI` and `CP-Gen` produce intervals that cover the ground truth policy value? **3)** Under what conditions do the `DR-PPI` estimates outperform baseline approaches?

### 5.1  Datasets

To answer our empirical questions, we use the following domains.

**Inventory Control** (Foffano et al., 2023): We adapt this simulator to accommodate a continuous state and reward space.

**Sepsis** (Oberst & Sontag, 2019): In this popular sepsis simulator, the goal is to successfully discharge a simulated patient. We approximate the dynamics using a feed-forward network.

**D4RL HalfCheetah** (Fu et al., 2020): The HalfCheetah environment is a Mujoco task in the D4RL suite where the goal is to get the cheetah to move forward. Here, we approximate the dynamics using a variational auto-encoder (VAE) (Gao et al., 2023).

**MIMIC-IV** (Johnson et al., 2024; Goldberger et al., 2000): We consider a subset of patients from MIMIC-IV that receive potassium repletion. To emulate a setting in which we have access to both a behavior and target cohort, we construct two sub-cohorts. The behavior sub-cohort consists of patients who receive lower dosages (<20 mEq/L), and the target sub-cohort consists of patients who receive higher dosages (>= 20mEq/L).

We use a VAE to generate synthetic trajectories. Our goal is to learn the value of the target policy (i.e., repletion strategy in the higher-dosage cohort).

The domains we consider range in complexity and relevance to real-world settings. This breadth allows us to understand how specific environmental factors, including simulator quality and the size of the state and action space, impact estimator performance.

## 5.2 Baselines

In addition to the baseline proposed in Foffano et al. (2023), we compare to the following approaches:
**Importance Sampling (IS)**: We use standard IS, deriving a bound using central limit theorem (CLT) or bootstrapping.
**Augmented Importance Sampling (AugIS)**: We use both the original dataset and a set of synthetic trajectories to calculate an IS estimate, with bounds estimated using either CLT or boostrapping.
**Direct Method (DM)**: We use rollouts from the learned model and calculate the expectation of the trajectory returns. DM estimates do not produce CIs.
**Doubly Robust (DR)**: Here, we compute a DR estimated using DQL to learn the reward model.
**Augmented Doubly Robust (AugDR)**: Here, we use both offline trajectories and synthetic trajectories to learn a Deep Q-learning (DQL) reward model and then compute a DR estimate.
**Q-Bootstrap**: Here, we fit a $Q$-function using the behavior dataset and use it to learn a bootstrapped estimate of $V^{\pi_e}(s)$.

| Setting | Ground truth $V^{\pi_e}(s)$ | DM (Point Estimate) | Foffano et al. Interval Length | Q-bootstrap Interval Length | CP-Gen Interval Length |
|---|---|---|---|---|---|
| Inventory | -412.85 | -120.57 | 8550.00 | 520.64* | **5531.60** |
| Sepsis | -0.40 | -0.12 | **1** | 0.02* | 1.90 |
| D4RL Half Cheetah | 1990.39 | 1393.98 | 190.00 | 60.00* | **40.07** |
| MIMIC-IV | 1 | 0.689 | 1.00 | 2.20 | **0.13** |

Table 2: **CP-Gen outperforms baselines across domains with continuous state-spaces**, producing conformal prediction intervals for policy value estimation. For methods that produce an interval, we report the interval length for $\alpha = 0.05$. The interval length that is shortest that also covers the ground truth policy value $V^{\pi_e}(s)$ is in **bold**. Intervals that do not cover the ground truth policy are marked with an asterisk (*).

| Setting | $V^{\pi_e}$ | IS (CLT) | IS (Bootstrap) | AugIS (CLT) | AugIS (Bootstrap) | DR (CLT) | AugDR (CLT) | DM | DR-PPI |
|---|---|---|---|---|---|---|---|---|---|
| Inventory | -428.51 | 1929.7 | 1982.14 | 54.66* | 49.68* | 2744.46 | 107.22* | -100.53 | **1918.38** |
| Sepsis | -0.56 | 1.58 | 1.48 | 0.008* | 0.007* | 1.23 | 1.272e+11 | -0.4 | **1.19** |
| D4RL Half Cheetah | 1975.75 | 281.88 | **271.67** | 151.93* | 142.14* | 5.514e+31 | 1.17e+32 | 1423.57 | 281.81 |
| MIMIC-IV | 0.746 | 1.19 | **1.09** | 0.008* | 0.007* | 7.566e+21 | 0.011* | 0.69 | 1.19 |

Table 3: **DR-PPI produces valid confidence intervals across all domains.** For methods that produce an interval, we report interval lengths for the same coverage ($\alpha = 0.05$), and **bold** the interval with the smallest size that also covers the ground truth policy value $V^{\pi_e}$. Intervals that do not cover the ground truth value of the policy are marked with an asterisk (*).

## 5.3 Results

**CP-Gen produces valid CP intervals**. As discussed in Section 3, to scale prior conformal prediction approaches to large MDPs, we use an $\epsilon$-approximation strategy. Despite this approximation, we find that **CP-Gen** still results in conformal prediction intervals that cover $V^{\pi_e}(s)$ with the specified confidence level, often with a smaller interval size than baseline approaches (Table 2, full intervals reported in Table 6). We compare to a DM-style baseline where we average the return of synthetic trajectories that start in the given initial state. We find that the DM baseline can produce a biased result with a poor generative model (e.g., in D4RL, MIMIC-IV). We also evaluate the baseline reported in Foffano et al. (2023). This baseline covers the ground truth value but produces wider intervals than **CP-Gen** in all environments with continuous state spaces (e.g., all environments except Sepsis). These results suggest that **CP-Gen** newly enables conformal prediction for OPE in MDP settings with large or continuous state and action spaces.

| Setting | Method | Coverage Rate | Average Length of Interval |
|---------|--------|:-------------:|:--------------------------:|
| Inventory | CP-Gen | 98% | 5576.85 |
| Inventory | DR-PPI | 96% | 1951.14 |
| Sepsis | CP-Gen | 92% | 1.442 |
| Sepsis | DR-PPI | 96% | 1.172 |
| D4RL Half Cheetah | CP-Gen | 92% | 64.77 |
| D4RL Half Cheetah | DR-PPI | 96% | 291.95 |
| MIMIC-IV | CP-Gen | 94% | 0.211 |
| MIMIC-IV | DR-PPI | 98% | 1.163 |

Table 4: **Empirical coverage rates across all domains for CP-Gen and DR-PPI.** We report coverage rates (out of 25 iterations for Half Cheetah, out of 50 iterations for the other three settings) corresponding to $\alpha = 0.05$.

**DR-PPI identifies valid confidence intervals that cover $V^{\pi_e}$ across all domains**. Across all domains, DR-PPI produces valid CIs that cover $V^{\pi_e}$, as our theoretical results suggest (Table 3). In contrast, most baseline approaches either have wide CIs or have intervals that do not cover $V^{\pi_e}$. In fact, any baseline approach that uses generated trajectories (e.g., AugIS, AugDR) produces a biased interval, which suggests that naively incorporating auxiliary synthetic trajectories results in a biased estimator. Furthermore, we find that in the D4RL and MIMIC-IV settings, DQL is unable to identify an accurate Q function; as a result, the CIs become exponentially large.

**DR-PPI performs best in stochastic domains with high quality generative models**. Finally, we clarify the settings in which DR-PPI outperforms baselines. When the environment is deterministic (e.g., D4RL HalfCheetah), or the generative model is of poor quality (e.g., MIMIC-IV) DR-PPI performs similarly to the IS baselines. In such settings we do not get a favorable variance reduction from the synthetic trajectories because they are often highly deterministic. In contrast, in settings where the environment is stochastic and our learned generative model is good (e.g., Inventory, Sepsis), DR-PPI has tighter CIs. Given that both IS with bootstrapping and DR-PPI produce valid CIs, we recommend a simple rule: use the estimator with the narrower interval. We defer a rigorous selection criterion to future work.

**Empirical coverage rates.** Finally, we investigate empirical coverage rates for both methods in the all settings (Table 4). We note that in all settings, DR-PPI covers the ground truth value of the policy, and that CP-Gen achieves the requested coverage in Inventory. We believe that the slight loss of coverage for CP-Gen in the Sepsis, D4RL, and MIMIC-IV settings is due to a higher $\Delta_w$. For example, the Sepsis environment, due to its discrete state and reward space, exhibits weak Lipschitz continuity, with a large Lipschitz constant. Furthermore, in this setting, $C_{ips}$, the upper bound of the IPS ratios, is large given that the target and behavior policies are quite distinct. As suggested in Theorem 1, these two factors contribute to a higher $\Delta_w$, which results in a small loss of coverage.

## 6 Conclusion

Here, we take steps toward uncertainty-aware OPE in settings that combine real and synthetic trajectories. We present two complementary approaches, CP-Gen and DR-PPI, that use auxiliary data to construct CIs for OPE. CP-Gen calculates state-conditioned policy values, while DR-PPI estimates unconditional policy values. We provide theoretical guarantees (Section 4) and examine behavior in four empirical domains(Section 5). Our results illustrate that obtaining valid CIs for OPE with auxiliary data is feasible across a variety of domains, from fully synthetic settings to real-world EHR data.

**Limitations and future work.** Our work considers settings with continuous state spaces of moderate dimension. When applied to higher-dimensional settings, the choice of $\epsilon_s, \epsilon_r$ becomes increasingly consequential as poor choices can inflate $\Delta_w$, thus reducing coverage rates. Guidance on selecting these parameters is provided in Section 3.3. Future work can explore developing more principled procedures for setting $\epsilon_s$ and $\epsilon_r$, alternative classes of generative models such as diffusion models, and investigate strategies to mitigate the

impact of poor-quality generated trajectories. More broadly, we see value in analyzing these approaches under distribution shift and partial observability.

**Broader Impact Statement**

In this work, we propose two strategies to estimate confidence intervals for off-policy evaluation when used with both real and synthetic data. We believe this work is foundational and acknowledge that it has the potential to improve the downstream application of RL policies in high-stakes domains.

**Acknowledgments**

The authors would like to thank Matthew Jörke, Shengpu Tang, John Duchi, and Alex Nam for feedback on early versions of this manuscript. AM was funded in part by a Stanford Data Science fellowship. BEE was funded in part by grants from the Parker Institute for Cancer Immunology (PICI), the Chan-Zuckerberg Institute (CZI), the Biswas Family Foundation, NIH NHGRI R01 HG012967, and NIH NHGRI R01 HG013736. BEE is a CIFAR Fellow in the Multiscale Human Program. BEE is on the Scientific Advisory Board for ArrePath Inc, GSK AI for Cancer, and Freenome; she consults for Neumora.

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

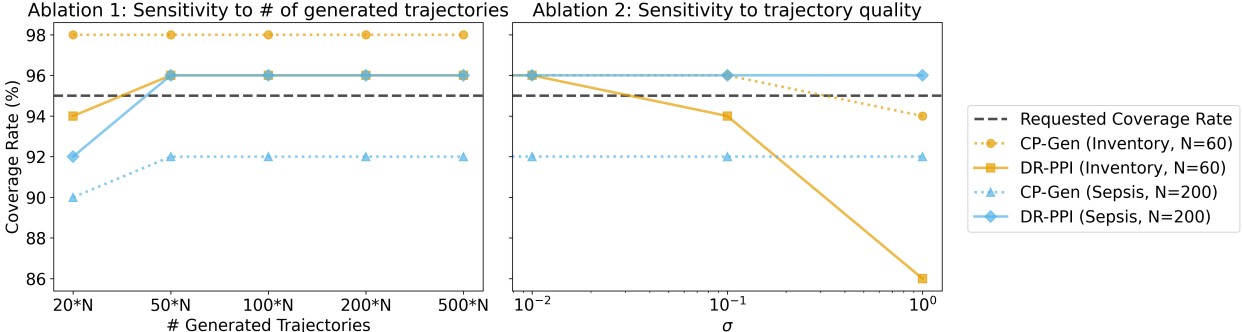

Figure 1: **DR-PPI and CP-Gen are robust to annotation quality and improve in quality as the number of generated trajectories increase** in the Inventory and Sepsis environments. (Left) We fix $N$ (i.e., number of behavior trajectories) for both the Inventory ($N = 60$) and Sepsis ($N = 200$) settings. We alter the number of generated trajectories from $20 * N$ to $500 * N$. We report the coverage rate across 50 iterations for $\alpha = 0.05$. (Right) We fix the number of generated trajectories at $100 * N$. We vary the quality of the generated trajectories by adding noise in the form of $\mathcal{N}(0, \sigma^2)$, similar to prior work (Laskin et al., 2020). We report coverage rate across 50 iterations, for $\alpha = 0.05$.

# A    Additional Empirical Results

First, we discuss additional empirical results. In the main text, we report **CP-Gen** and **DR-PPI** results across all domains. First, we include results in the Inventory setting that investigate the performance of our methods if the behavior policy $\pi_b$ is unknown. We report coverage rates across 50 trials, and define the estimation error in terms of $\epsilon_x$. Here, the true behavior policy is static and uniform across all actions [1/11 for - in range(11)]. The estimated behavior policy is defined as $[1/11 + \epsilon_x$ for - in range(5)$] + [1/11 - \epsilon_x$ for - in range(5)$] + [1/11]$. That is, the perturbation by increases the probability of the first 5 actions, decreases the probability of the next 5 actions, and retains the probability of the last action. Our results (Table 5) support the claim that our methods are robust to small errors in the estimation of the behavior policy, with coverage dropping at $\epsilon_x \geq 0.01$. We observe that the drop in coverage is more pronounced for **DR-PPI**, though we also note that an $\epsilon_x$ of 0.03 refers to a very large degree of misspecification.

| Method | $\epsilon_x = 0$ | $\epsilon_x = 0.005$ | $\epsilon_x = 0.01$ | $\epsilon_x = 0.02$ | $\epsilon_x = 0.03$ |
|---|---|---|---|---|---|
| **CP-Gen** | 98% | 92% | 84% | 84% | 80% |
| **DR-PPI** | 96% | 94% | 90% | 82% | 72% |

Table 5: **CP-Gen and DR-PPI are robust to moderate levels of misspecification in $\pi_b$.** Here, we study the Inventory setting and report coverage rates (out of 50) as we vary $\epsilon_x$, the degree to which the behavior policy $\pi_b$ is perturbed. We request a coverage corresponding to $\alpha = 0.05$.

Additionally, we discuss the performance of our estimators as a function of the number of generated synthetic trajectories and the noise of the synthetic trajectories (Figure 1). We find that in the majority of settings, **DR-PPI** and **CP-Gen** achieve the requested coverage ($\alpha = 0.05$). When **DR-PPI** does not achieve the requested coverage, we believe there are too few generated trajectories (i.e., non-asymptotic result). When **CP-Gen** does not achieve the requested coverage, we believe this is due to a higher $\Delta_w$ (similar to the empirical coverage experiment in Table 4). We also study the effect of the trajectory quality on the performance of our estimators. For the Inventory setting, we find that the coverage rate only slightly decreases at higher degrees of noise. For the sepsis setting, we find that coverage does not change in comparison to perfect generated trajectories, which suggests that our methods can correct for noisy synthetic trajectories in these settings.

| Setting | $V^{\pi e}(s)$ | DM | Foffano et al. | | Q-bootstrap | | CP-Gen | |
|---|---|---|---|---|---|---|---|---|
| | | | Interval | Covers? | Interval | Covers? | Interval | Covers? |
| Inventory | -412.85 | -120.57 | (-6040, 2510) | ✓ | (-1566.32, -1045.68) | ✗ | **(-4449.27, 1082.33)** | ✓ |
| Sepsis | -0.40 | -0.12 | **(-1,0)** | ✓ | (-0.01, 0.01) | ✗ | (-1.36, 0.54) | ✓ |
| D4RL Half Cheetah | 1990.39 | 1393.98 | (1750, 1940) | ✓ | (1820, 1880) | ✗ | **(1964.35, 2004.42)** | ✓ |
| MIMIC-IV | 1 | 0.689 | (0,1) | ✓ | (-1.28, 0.92) | ✓ | **(0.977, 1.1012)** | ✓ |

Table 6: **CP-Gen outperforms baselines across domains with continuous state-spaces**, producing conformal prediction intervals that cover the true policy value, $V^{\pi_e}(s)$. For methods that produce an interval, we report the interval for $\alpha = 0.05$ and whether the interval covers the true policy value. The method with the smallest interval length that covers the ground truth policy value is bolded.

| Setting | $V^{\pi e}$ | IS (CLT) | IS (Bootstrap) | AugIS (CLT) | AugIS (Bootstrap) | DR (CLT) | AugDR (CLT) | DM | DR-PPI |
|---|---|---|---|---|---|---|---|---|---|
| Inventory | -428.51 | (-2139.27, -209.57) | (-2209.73, -227.59) | (-808.47, -753.81) | (-806.41, -756.73) | (-1804.16, 940.30) | (-914.72, -807.50) | -100.53 | **(-2106.27, -187.89)** |
| Sepsis | -0.56 | (-1.68, -0.10) | (-1.73, -0.25) | (-0.002, 0.006) | (-0.001, 0.006) | (-1.67, -0.44) | (-2.92e+10, 9.8e+10) | -0.4 | **(-1.45, -0.26)** |
| D4RL Half Cheetah | 1975.75 | (1814.37, 2096.25) | **(1802.37, 2074.04)** | (970.46, 1122.39) | (973.22, 1115.36) | (-1.320e+32, 4.194e+31) | (-3.59e+31, 7.58e+30) | 1423.57 | (1820.79, 2102.60) |
| MIMIC-IV | 0.746 | (0.31, 1.50) | **(0.56, 1.65)** | (0.711, 0.719) | (0.711, 0.718) | (-5.874e+21, 1.892e+21) | (0.719, 0.730) | 0.69 | (0.29, 1.48) |

Table 7: **DR-PPI produces valid confidence intervals across all domains.** We report all CIs for the same coverage ($\alpha = 0.05$), and bold the interval with the smallest size that also covers the ground truth policy value $V^{\pi_e}$.

## B  Code and Synthetic Trajectory Generation Procedure

The code to reproduce all experiments is provided on Github.

Synthetic trajectories are generated by rolling out a learned dynamics model under a specified policy. For the Inventory and Sepsis settings, we use a feedforward neural network for the dynamics model, and for the D4RL and MIMIC settings, we use a VAE. The feedforward neural network has 2 hidden layers trained on behavior data to predict the next state and reward for the dynamics model.

For the D4RL and MIMIC-IV settings, we use a VAE. We adopt a training procedure similar to (Gao et al., 2023). In particular, the VAE encodes consecutive observations into a 16-dimensional latent space via a two-layer MLP and an LSTM (hidden size 64), then decodes using an ensemble of 10 branches, each with independently parameterized MLPs for state reconstruction, reward prediction, and episode termination. Final predictions are a learned weighted average across branches. The model is trained by maximizing an ELBO that balances state/reward/termination likelihoods against KL regularization of the latent dynamics, with an additional loss aligning encoder and decoder LSTM hidden states. At inference time, we sample an initial state from the empirical state distribution, sample an action according to the target policy, and iteratively sample next states from the learned dynamics model for a fixed horizon of H steps.

## C  Empirical Settings

In the main text, we consider empirical results using four datasets. Here, we expand the description of each dataset.

**Inventory Control**:
The inventory control simulator is adapted from a version featured in Foffano et al. (2023). The state is the current inventory, actions are the number of units purchased, and reward is the end-of-day earnings. We make several adaptations to make this inventory control environment more suitable for our work. First, the distribution of the stochastic demand in the inventory is $o$ is changed from Poisson to normal $\mathcal{N}(\mu, \sigma)$. Additionally, the cost of buying items is $k \times \mathbb{1}_{\{a>0\}} + c(\min(N, x_t + a) - x_t)$, where $k > 0$ is the fixed cost for a single order, $c > 0$ is the cost of a single unit bought, $N$ is the inventory upper-bound, $x_t$ is

the state at timestep $t$, and $a_t$ is the action at timestep $t$. The next state $x_{t+1}$ is calculated as $x_{t+1} = \max(0, \min(N, x_t + a_t) - o_{t+1})$. The instantaneous reward observed at the end of the day is the sum of the costs and earnings listed above (e.g., $r(x_t, a_t, x_{t+1}) = 10^2 \times (-k1_{\{a_t > 0\}} - zx_t - c(\min(N, x_t + a_t) - x_t) + p\max(0, \min(N, x_t + a_t) - x_{t+1})))$. When testing our algorithm, we chose the following parameters: $N = 10, k = 1, c = 2, z = 2, p = 4, \mu = 5, \sigma = 10, H = 20$. We approximate the dynamics in this setting using a feed-forward network.

**Sepsis**:

The Sepsis simulator is taken directly from Oberst & Sontag (2019), which models a synthetic Sepsis treatment setting. The state is an 8-dimensional vector which contains information about relevant vitals and labs, indicators of ongoing treatment (e.g., antibiotics, vasopressors, ventilator), and an indication of whether the patient is diabetic. There are 8 possible actions, each corresponding to a different combination of 3 binary treatments (e.g., antibiotics, vasopressors, ventilator). The reward is $+1$ if the synthetic patient is off of treatment and has stable vitals, $-1$ if the patient has unstable vitals, and $+0$ otherwise. We do not alter any environment details, and report results with a maximum horizon of $H = 20$.

**D4RL HalfCheetah**:

The HalfCheetah environment is a Mujoco task in the D4RL suite (Fu et al., 2020). The cheetah is a two-dimensional robot that has 9 body parts and 8 joints connecting the body parts. Each state is represented as a 17-dimensional vector that contains information about the position and velocity of each of the joints. Each action is represented as a 5-dimensional vector, and applies torque to a subset of the joints, and the goal of the environment is to get the cheetah to move forward as quickly as possible. The reward corresponds to how far the cheetah traveled, with negative reward indicating that the cheetah moved backward. We report results using a maximum horizon of $H = 1000$.

**MIMIC-IV**:

MIMIC-IV is an electronic health records (EHR) dataset collected from patients admitted to the Beth Israel Deaconess Medical Center in Boston, MA (Johnson et al., 2024; Goldberger et al., 2000). We consider a subset of patients that receive potassium repletion through an intravenous (IV) line. We represent the patient state as a 20-dimensional vector containing information about important labs, administered medicines, and static covariates such as age and gender; each state represents a 4-hour interval in a patient's hospital stay. There are five actions, each corresponding to a dosage of potassium delivered through an IV. The reward function is a binary indicator of whether the patient's potassium lab value is within the potassium reference range (3.5-4.5 mmol/L), within 2 hours of receiving the administered potassium.

In the OPE task, we assume access to a behavior policy $\pi_b$ and a target policy $\pi_e$, and evaluate using RMSE. It is not immediately obvious how to create this setup within MIMIC-IV. To emulate a setting in which we have access to both a behavior and target cohort, we construct two sub-cohorts from the patients that receive potassium repletion. The behavior sub-cohort consists of patients who receive lower dosages ($<20$ mEq/L) of potassium, and the target sub-cohort consists of the patients who receive higher dosages ($>= 20$mEq/L) of potassium. The behavior policy corresponds to the repletion policy in the behavior cohort, and the target policy corresponds to the repletion policy in the target cohort. Both policies are inferred using behavior cloning. Our goal is to learn the value of the target policy, and a ground truth calculation of this value is calculated by averaging the returns of the target trajectories. We observe that the maximum horizon length is $H = 189$, though most patients have trajectories that are less than 20 timesteps.

## D   Computational cost of experiments

All experiments were conducted on an internally hosted cluster equipped with an NVIDIA RTX A6000 GPU featuring 48 GB of memory. In total, our experiments consumed approximately 250 compute hours, primarily driven by VAE training and Q-function learning on large datasets.

# E    Proofs for Theoretical Results

## E.1    Proof of Equation (9)

*Proof.*

$$w(s, \delta_{rr'}) := \frac{\mathbb{P}^{\pi_e}_{(S, \Delta_{rr'})}(s, \delta_{rr'})}{\mathbb{P}^{\pi_b}_{(S, \Delta_{rr'})}(s, \delta_{rr'})} \tag{25}$$

$$= \iint \frac{\mathbb{P}^{\pi_e}_{(S, \Delta_{rr'})}(s, \delta_{rr'})}{\mathbb{P}^{\pi_b}_{(S, \Delta_{rr'})}(s, \delta rr')} \frac{\mathbb{P}^{\pi_b}_{\tau, \tilde{\tau}|S, \Delta_{rr'}}(\tau, \tilde{\tau}|s, \delta_{rr'})}{\mathbb{P}^{\pi_b}_{\tau, \tilde{\tau}|S, \Delta_{rr'}}(\tau, \tilde{\tau}|s, \delta_{rr'})} \mathbb{P}^{\pi_e}_{\tau, \tilde{\tau}|S, \Delta_{rr'}}(\tau, \tilde{\tau}|s, \delta_{rr'}) d\tau d\tilde{\tau} \tag{26}$$

$$= \iint \frac{\mathbb{P}^{\pi_e}_{(S, \Delta_{rr'}, \tau, \tilde{\tau})}(s, \delta_{rr'}, \tau, \tilde{\tau})}{\mathbb{P}^{\pi_b}_{(S, \Delta_{rr'}, \tau, \tilde{\tau})}(s, \delta_{rr'}, \tau, \tilde{\tau})} \mathbb{P}^{\pi_b}_{\tau, \tilde{\tau}|S, \Delta_{rr'}}(\tau, \tilde{\tau}|s, \delta_{rr'}) d\tau d\tilde{\tau} \tag{27}$$

$$= \mathbb{E}_{\tau \sim p^{\pi_b}, \tilde{\tau} \sim \tilde{p}^{\pi_b}|S=s, \Delta_{rr'}=\delta_{rr'}} \left[ \frac{\mathbb{P}^{\pi_e}_{(S, \Delta_{rr'}, \tau, \tilde{\tau})}(s, \delta_{rr'}, \tau, \tilde{\tau})}{\mathbb{P}^{\pi_b}_{(S, \Delta_{rr'}, \tau, \tilde{\tau})}(s, \delta_{rr'}, \tau, \tilde{\tau})} \right] \tag{28}$$

$$= \mathbb{E}_{\tau \sim p^{\pi_b}, \tilde{\tau} \sim \tilde{p}^{\pi_b}|S=s, \Delta_{rr'}=\delta_{rr'}} \left[ \frac{P(\delta_{rr'}|s, \tau, \tilde{\tau}) P^{\pi_e}(\tau|s) \tilde{P}^{\pi_e}(\tilde{\tau}|s)}{P(\delta_{rr'}|s, \tau, \tilde{\tau}) P^{\pi_b}(\tau|s) \tilde{P}^{\pi_b}(\tilde{\tau}|s)} \right] \tag{29}$$

$$= \mathbb{E}_{\tau \sim p^{\pi_b}, \tilde{\tau} \sim \tilde{p}^{\pi_b}|S=s, \Delta_{rr'}=\delta_{rr'}} \left[ \frac{P^{\pi_e}(\tau|s) \tilde{P}^{\pi_e}(\tilde{\tau}|s)}{P^{\pi_b}(\tau|s) \tilde{P}^{\pi_b}(\tilde{\tau}|s)} \right] \tag{30}$$

$$= \mathbb{E}_{\tau \sim p^{\pi_b}, \tilde{\tau} \sim \tilde{p}^{\pi_b}|S=s, \Delta_{rr'}=\delta_{rr'}} \left[ \frac{\prod_{t=1}^{H} \pi_e(a_t|s_t) p(s_{t+1}|s_t, a_t) \pi_e(\tilde{a}_t|\tilde{s}_t) \tilde{p}(\tilde{s}_{t+1}|\tilde{s}_t, \tilde{a}_t)}{\prod_{t=1}^{H} \pi_b(a_t|s_t) p(s_{t+1}|s_t, a_t) \pi_b(\tilde{a}_t|\tilde{s}_t) \tilde{p}(\tilde{s}_{t+1}|\tilde{s}_t, \tilde{a}_t)} \right] \tag{31}$$

$$= \mathbb{E}_{\tau \sim p^{\pi_b}, \tilde{\tau} \sim \tilde{p}^{\pi_b}|S=s, \Delta_{rr'}=\delta_{rr'}} \left[ \frac{\prod_{t=1}^{H} \pi_e(a_t|s_t) \pi_e(\tilde{a}_t|\tilde{s}_t)}{\prod_{t=1}^{H} \pi_b(a_t|s_t) \pi_b(\tilde{a}_t|\tilde{s}_t)} \right] \tag{32}$$

$\square$

## E.2    Additional Assumptions

First, we formally state the assumptions used in prior literature (Farajtabar et al., 2018; Thomas & Brunskill, 2016) to support our theoretical results.

**Assumption 4** (Common support). $\pi_e(a|s) > 0 \rightarrow \pi_b(a|s) > 0, \forall s \in \mathcal{S}, \forall a \in \mathcal{A}.$

**Assumption 5** (Bounded return). $0 \leq J(\tau) \leq C_r$ for all $\tau \sim p.$

**Assumption 6** (Bounded IPS weights). $c_{ips} \leq \frac{\pi_e(a|s)}{\pi_b(a|s)} \leq C_{ips}, \forall s \in \mathcal{S}, \forall a \in \mathcal{A}.$

These assumptions are standard in the literature and minimally restrictive, thus enabling the analysis of `CP-Gen`'s performance under realistic conditions. We also consider Assumption 7, a mild regularity condition that holds in a wide variety of real-world MDPs, including those with heterogeneous populations and varied outcomes, such as clinical settings with diverse patient cohorts. Prior work in bandit and reinforcement learning has used similar assumptions (Qian & Yang, 2016; Bastani et al., 2021; Neu et al., 2010).

**Assumption 7** (Bounded density). The joint density of $(S, \Delta_{rr'})$ under $\mathbb{P}^{\pi_b}$ is uniformly bounded: $p_{\min} \leq p(s, \delta_{rr'}) \leq p_{\max}, \forall s, \delta_{rr'}.$

## E.3    Proof of Theorem 1

**Lemma 3** (Coupling Lemma). *Let $X$ and $Y$ be random variables with probability distributions $\mu$ and $\nu$ over $\Omega$. There always exists a coupling $w$ on $\Omega \times \Omega$ s.t.,*

$$\|\mu - \nu\|_{TV} = P(X \neq Y).$$

*Proof.* This is a prior known result. Reference includes Daskalakis et al. (2011).

$\square$

**Lemma 4.** *Assume the action space is bounded, $\|a\| \leq C_a$. Given two states $s, s_1$, there exists an optimal coupling, such that*

$$\mathbb{E}_{a\sim\pi(\cdot|s),a_1\sim\pi(\cdot|s_1)}\|a - a_1\| \leq 2C_a P^\pi(a \neq a_1) = 2C_a TV(\pi(\cdot|s), \pi(\cdot|s_1)) \leq 2C_a L_\pi \|s - s_1\|. \tag{33}$$

*Proof.* This is a direct consequence of Coupling Lemma. $\qquad\square$

**Lemma 5.** *Assume the action space is bounded, $\|s\| \leq C_s$. Given two states $s_{t-1}, s'_{t-1}$, there exists an optimal coupling, such that*

$$\mathbb{E}_{a_{t-1}\sim\pi(\cdot|s_{t-1}),a'_{t-1}\sim\pi(\cdot|s'_{t-1}),s_t\sim p(\cdot|s_{t-1},a_{t-1}),s'_t\sim p(\cdot|s'_{t-1},a'_{t-1})}\|s_t - s'_t\| \tag{34}$$

$$\leq 2C_s P^\pi(s_t \neq s'_t) \tag{35}$$

$$= 2C_s \mathbb{E}_{a_{t-1}\sim\pi(\cdot|s_{t-1}),a'_{t-1}\sim\pi(\cdot|s'_{t-1})} TV(p(\cdot|s_{t-1},a_{t-1}), p(\cdot|s'_{t-1},a'_{t-1})) \tag{36}$$

$$\leq 2C_s (L_{p,s}\|s_{t-1} - s'_{t-1}\| \tag{37}$$

$$+ L_{p,a}\mathbb{E}_{a_{t-1}\sim\pi(\cdot|s_{t-1}),a'_{t-1}\sim\pi(\cdot|s'_{t-1})}\|a_{t-1} - a'_{t-1}\|) \tag{38}$$

$$\leq 2C_s (L_{p,s} + 2C_a L_\pi L_{p,a})\|s_{t-1} - s'_{t-1}\|. \tag{39}$$

*Thus, if $\|s_1 - s'_1\| \leq \epsilon_s$, then*

$$\mathbb{E}_{\tau,\tau'\sim p^\pi}\|s_t - s'_t\| \leq L^{t-1}\mathbb{E}_{\tau,\tau'\sim p^\pi}\|s_1 - s'_1\| \leq \epsilon_s L^{t-1}, \tag{40}$$

*where $L = 2C_s(L_{p,s} + 2C_a L_\pi L_{p,a})$.*

*And the same holds also for $\tilde{p}$.*

*Proof.* This is a direct consequence of Coupling Lemma. $\qquad\square$

**Lemma 6.** *$\forall s, a, s', a', s_1, a_1, s'_1, a'_1$, for $\pi \in \{\pi_b, \pi_e\}$,*

$$|\pi(a|s)\pi(a'|s') - \pi(a_1|s_1)\pi(a'_1|s'_1)| \leq L_{\pi,s}(\|s - s_1\| + \|s' - s'_1\|) + L_{\pi,a}(\|a - a_1\| + \|a' - a'_1\|). \tag{41}$$

*Proof.*

$$|\pi(a|s)\pi(a'|s') - \pi(a_1|s_1)\pi(a'_1|s'_1)| \tag{42}$$

$$\leq |\pi(a|s)\pi(a'|s') - \pi(a|s)\pi(a'_1|s'_1)| + |\pi(a|s)\pi(a'_1|s'_1) - \pi(a_1|s_1)\pi(a'_1|s'_1)| \tag{43}$$

$$\leq L_{\pi,s}\|s' - s'_1\| + L_{\pi,a}\|a' - a'_1\| + L_{\pi,s}\|s - s_1\| + L_{\pi,a}\|a - a_1\|. \tag{44}$$

$$\square$$

**Lemma 7.** *Assume $\forall s, a$, for $\pi \in \{\pi_b, \pi_e\}$, $\pi(a|s) \geq c > 0$. Define the per-step importance-ratio*

$$f(s, a, s', a') = \frac{\pi_e(a|s)\pi_e(a'|s')}{\pi_b(a|s)\pi_b(a'|s')},$$

*we can derive that there is a constant $L_f(c, c_{ips}, C_{ips}, L_{\pi,s}, L_{\pi,a})$ such that*

$$|f(s, a, s', a') - f(s_1, a_1, s'_1, a'_1)| \leq L_f(\|s - s_1\| + \|s' - s'_1\| + \|a - a_1\| + \|a' - a'_1\|). \tag{45}$$

*Proof.*

$$|f(s, a, s', a') - f(s_1, a_1, s'_1, a'_1)| = \frac{|\pi_e(a|s)\pi_e(a'|s')\pi_b(a_1|s_1)\pi_b(a'_1|s'_1) - \pi_e(a_1|s_1)\pi_e(a_1|s'_1)\pi_b(a|s)\pi_b(a'|s')|}{\pi_b(a|s)\pi_b(a'|s')\pi_b(a_1|s_1)\pi_b(a'_1|s'_1)}$$

(46)

$$\leq \frac{|\pi_e(a|s)\pi_e(a'|s')\pi_b(a_1|s_1)\pi_b(a'_1|s'_1) - \pi_e(a|s)\pi_e(a'|s')\pi_b(a|s)\pi_b(a'|s')|}{\pi_b(a|s)\pi_b(a'|s')\pi_b(a_1|s_1)\pi_b(a'_1|s'_1)}$$

(47)

$$+ \frac{|\pi_e(a|s)\pi_e(a'|s')\pi_b(a|s)\pi_b(a'|s') - \pi_e(a_1|s_1)\pi_e(a_1|s'_1)\pi_b(a|s)\pi_b(a'|s')|}{\pi_b(a|s)\pi_b(a'|s')\pi_b(a_1|s_1)\pi_b(a'_1|s'_1)}$$

(48)

$$\leq 2c^4(L_{\pi,s}\|s' - s'_1\| + L_{\pi,a}\|a' - a'_1\| + L_{\pi,s}\|s - s_1\| + L_{\pi,a}\|a - a_1\|) \quad (49)$$

$$\leq L_f(\|s - s_1\| + \|s' - s'_1\| + \|a - a_1\| + \|a' - a'_1\|). \quad (50)$$

$$\square$$

**Theorem 8** ($\epsilon-$approximation Error Bound)**.**

$$|w_\epsilon(s, \delta_{rr'}) - w(s, \delta_{rr'})| \leq L_s\epsilon_s + L_r\epsilon_r,$$

*where*

$$L_s = 2C_{ips}^{2(H-1)}(2C_aL_\pi + 1)L_f\frac{L^H - 1}{L - 1}$$

*Proof.* Define

$$g(\tau, \tau') = \prod_{t=1}^{H} f(s_t, a_t, s'_t, a'_t).$$

By the telescoping-product bound and the Lipschitz of each $f$,

$$|g(\tau, \tau') - g(\tau_1, \tau'_1)| = |\prod_{t=1}^{H} f(s_t, a_t, s'_t, a'_t) - \prod_{t=1}^{H} f(s_{1,t}, a_{1,t}, s'_{1,t}, a'_{1,t})| \quad (51)$$

$$= |\prod_{t=1}^{H} f(s_t, a_t, s'_t, a'_t) - \prod_{t=1}^{H-1} f(s_t, a_t, s'_t, a'_t)f(s_{1,H}, a_{1,H}, s'_{1,H}, a'_{1,H}) \quad (52)$$

$$+ \prod_{t=1}^{H-1} f(s_t, a_t, s'_t, a'_t)f(s_{1,H}, a_{1,H}, s'_{1,H}, a'_{1,H}) - \prod_{t=1}^{H-2} f(s_t, a_t, s'_t, a'_t)\prod_{t=H-1}^{H} f(s_{1,t}, a_{1,t}, s'_{1,t}, a'_{1,t}) \quad (53)$$

$$+ \cdots - \prod_{t=1}^{H} f(s_{1,t}, a_{1,t}, s'_{1,t}, a'_{1,t})| \quad (54)$$

$$\leq \sum_{t=1}^{H}(\prod_{i \neq t} C_{ips}^2)|f(s_t, a_t, s'_t, a'_t) - f(s_{1,t}, a_{1,t}, s'_{1,t}, a'_{1,t})| \quad (55)$$

$$\leq C_{ips}^{2(H-1)}L_f\sum_{t=1}^{H}(\|s_t - s_{1,t}\| + \|a_t - a_{1,t}\| + \|s'_t - s'_{1,t}\| + \|a'_t - a'_{1,t}\|). \quad (56)$$

Taking expectations under the optimal coupling gives

$$|w(s, \delta_{rr'}) - w(s', \delta_{rr'})| = |\mathbb{E}_{\tau \sim p^{\pi_b}, \tau' \sim \tilde{p}^{\pi_b}}[g(\tau, \tau') \mid s] - \mathbb{E}_{\tau_1 \sim p^{\pi_b}, \tau'_1 \sim \tilde{p}^{\pi_b}}[g(\tau_1, \tau'_1) \mid s']| \tag{57}$$

$$\leq \mathbb{E}_{\tau, \tau_1 \sim p^{\pi_b}, \tau', \tau'_1 \sim \tilde{p}^{\pi_b}}[|g(\tau, \tau') - g(\tau_1, \tau'_1)| \mid s, s'] \tag{58}$$

$$= C_{ips}^{2(H-1)} L_f \sum_{t=1}^{H} \mathbb{E}_{\tau, \tau_1 \sim p^{\pi_b}, \tau', \tau'_1 \sim \tilde{p}^{\pi_b}}(\|s_t - s_{1,t}\| + \|a_t - a_{1,t}\| + \|s'_t - s'_{1,t}\| + \|a'_t - a'_{1,t}\|) \tag{59}$$

$$\leq C_{ips}^{2(H-1)} L_f \sum_{t=1}^{H} 2(2C_a L_\pi + 1)\epsilon_s L^{t-1} \tag{60}$$

$$= L_s \epsilon_s \tag{61}$$

Hence $x \mapsto w(x, y)$ is $L_s$-Lipschitz. Finally, notice that

$$w_\epsilon(s, \delta_{rr'}) = \mathbb{E}^{\pi_b}\big[w(S, \Delta_{rr'}) \mid S \in B(s, \epsilon_s), \Delta_{rr'} \in B(\delta_{rr'}, \epsilon_r)\big],$$

so

$$\big|w_\epsilon(s, \delta_{rr'}) - w(s, \delta_{rr'})\big| = \mathbb{E}^{\pi_b}\big[w(S, \Delta_{rr'}) - w(s, \delta_{rr'}) \mid S \in B(s, \epsilon_s), \Delta_{rr'} \in B(\delta_{rr'}, \epsilon_r)\big] \tag{62}$$

$$\leq \sup_{\|s'-s\| \leq \epsilon_s, |\delta'_{rr'} - \delta_{rr'}| \leq \epsilon_r} \big|w(s', \delta'_{rr'}) - w(s, \delta_{rr'})\big| \tag{63}$$

$$\leq L_s \epsilon_s + L_r \epsilon_r. \tag{64}$$

as claimed. □

**Lemma 9.** *There exists an* $(\epsilon_s^0, \epsilon_r^0)-$*covering of* $S \times \Delta_{rr'}$*, denoted as* $\mathcal{N} = \{(s_i, \delta_{rr',j})\}_{i=1,\ldots,N_S(\epsilon_s^0); j=1,\ldots,N_{\Delta_{rr'}}(\epsilon_r^0)}$*, such that with probability* $\geq 1 - \delta$*,*

$$|\hat{w}_\epsilon(s_i, \delta_{rr',j}) - w_\epsilon(s_i, \delta_{rr',j})| \leq (C_{ips}^2 - c_{ips}^2)\sqrt{\frac{\ln(2N_S(\epsilon_s^0)N_{\Delta_{rr'}}(\epsilon_r^0)/\delta)}{N_{\min}}}, \forall i, j, \tag{65}$$

*where* $N_{\min} = np_{\min}Vol(B(\epsilon_s, \epsilon_r))$.

*Proof.* Fix a covering point $(s_i, \delta_{rr',j}) \in \mathcal{N}$. Recall that

$$w_\epsilon(s_i, \delta_{rr',j}) = \mathbb{E}[f(\tau, \tilde{\tau}) \mid s_0 \in B(s_i, \epsilon_s), \; \Delta_{rr'} \in B(\delta_{rr',j}, \epsilon_r)], \tag{66}$$

where

$$f(\tau, \tilde{\tau}) := \frac{\prod_{t=1}^{H} \pi_e(a_t|s_t)\pi_e(\tilde{a}_t|\tilde{s}_t)}{\prod_{t=1}^{H} \pi_b(a_t|s_t)\pi_b(\tilde{a}_t|\tilde{s}_t)}. \tag{67}$$

By the bounded IPS assumption, we have

$$c_{ips}^2 \leq f(\tau, \tilde{\tau}) \leq C_{ips}^2 \tag{68}$$

almost surely.

For each $(i, j)$, let

$$\mathcal{I}_{i,j} := \{k \in [n] : s_{0,k} \in B(s_i, \epsilon_s), \; \Delta_{rr',k} \in B(\delta_{rr',j}, \epsilon_r)\} \tag{69}$$

denote the set of samples falling into the corresponding $(\epsilon_s, \epsilon_r)$-ball, and let

$$N_{i,j} := |\mathcal{I}_{i,j}|. \tag{70}$$

Then the empirical estimator can be written as

$$\hat{w}_\epsilon(s_i, \delta_{rr',j}) = \frac{1}{N_{i,j}} \sum_{k \in \mathcal{I}_{i,j}} f_k, \tag{71}$$

where $f_k := f(\tau_k, \tilde{\tau}_k)$.

Conditioned on the event $\mathcal{I}_{i,j}$, the random variables $\{f_k\}_{k \in \mathcal{I}_{i,j}}$ are i.i.d. and bounded in $[c_{ips}^2, C_{ips}^2]$, with mean

$$\mathbb{E}[f_k \mid k \in \mathcal{I}_{i,j}] = w_\epsilon(s_i, \delta_{rr',j}). \tag{72}$$

Therefore, by Hoeffding's inequality, for any $t > 0$,

$$\mathbb{P}\big(|\hat{w}_\epsilon(s_i, \delta_{rr',j}) - w_\epsilon(s_i, \delta_{rr',j})| > t \mid N_{i,j}\big) \leq 2 \exp\left(-\frac{2N_{i,j}t^2}{(C_{ips}^2 - c_{ips}^2)^2}\right). \tag{73}$$

Now suppose that the local sample size is uniformly lower bounded over the cover:

$$N_{i,j} \geq N_{\min}, \qquad \forall(i,j). \tag{74}$$

Then for every $(i,j)$,

$$\mathbb{P}\big(|\hat{w}_\epsilon(s_i, \delta_{rr',j}) - w_\epsilon(s_i, \delta_{rr',j})| > t\big) \leq 2 \exp\left(-\frac{2N_{\min}t^2}{(C_{ips}^2 - c_{ips}^2)^2}\right). \tag{75}$$

Applying a union bound over all $N_S(\epsilon_s^0) N_{\Delta_{rr'}}(\epsilon_r^0)$ points in the covering $\mathcal{N}$ yields

$$\mathbb{P}\big(\exists(i,j) : |\hat{w}_\epsilon(s_i, \delta_{rr',j}) - w_\epsilon(s_i, \delta_{rr',j})| > t\big) \tag{76}$$

$$\leq 2 N_S(\epsilon_s^0) N_{\Delta_{rr'}}(\epsilon_r^0) \exp\left(-\frac{2N_{\min}t^2}{(C_{ips}^2 - c_{ips}^2)^2}\right). \tag{77}$$

Setting the right-hand side equal to $\delta$ and solving for $t$ gives

$$t = (C_{ips}^2 - c_{ips}^2)\sqrt{\frac{\ln\big(2N_S(\epsilon_s^0) N_{\Delta_{rr'}}(\epsilon_r^0)/\delta\big)}{2N_{\min}}}. \tag{78}$$

Absorbing the factor of $1/\sqrt{2}$ into constants yields the stated bound:

$$|\hat{w}_\epsilon(s_i, \delta_{rr',j}) - w_\epsilon(s_i, \delta_{rr',j})| \leq (C_{ips}^2 - c_{ips}^2)\sqrt{\frac{\ln\big(2N_S(\epsilon_s^0) N_{\Delta_{rr'}}(\epsilon_r^0)/\delta\big)}{N_{\min}}}, \qquad \forall i, j, \tag{79}$$

with probability at least $1 - \delta$.

Finally, by definition of the minimum local mass $p_{\min}$ over the cover, each ball has probability at least

$$p_{\min} \text{Vol}(B(\epsilon_s, \epsilon_r)), \tag{80}$$

so the expected number of samples in each ball is at least

$$N_{\min} = n\, p_{\min} \text{Vol}(B(\epsilon_s, \epsilon_r)). \tag{81}$$

This proves the lemma. □

We now start the main proof of Theorem 1.

*Proof.*

$$\mathbb{E}^{\pi_b}|\hat{w}_\epsilon(S, \Delta_{rr'}) - w(S, \Delta_{rr'})| \leq \mathbb{E}^{\pi_b}|\hat{w}_\epsilon(S, \Delta_{rr'}) - w_\epsilon(S, \Delta_{rr'})| + \mathbb{E}^{\pi_b}|w_\epsilon(S, \Delta_{rr'}) - w(S, \Delta_{rr'})| \tag{82}$$

$$\leq \mathbb{E}^{\pi_b}|\hat{w}_\epsilon(S, \Delta_{rr'}) - w_\epsilon(S, \Delta_{rr'})| + L_s\epsilon_s + L_r\epsilon_r \tag{83}$$

For each $(s, \delta_{rr'})$, $\exists (s_i, \delta_{rr',j}) \in \mathcal{N}$, such that $\|s - s_i\| \leq \epsilon_s^0, \|\delta_{rr'} - \delta_{rr',j}\| \leq \epsilon_r^0$, so

$$\mathbb{E}^{\pi_b}|\hat{w}_\epsilon(S, \Delta_{rr'}) - w_\epsilon(S, \Delta_{rr'})| \leq \mathbb{E}^{\pi_b}[\underbrace{|\hat{w}_\epsilon(S, \Delta_{rr'}) - \hat{w}_\epsilon(s_i, \delta_{rr',j})|}_{1} + \underbrace{|\hat{w}_\epsilon(s_i, \delta_{rr',j}) - w_\epsilon(s_i, \delta_{rr',j})|}_{2} \tag{84}$$

$$+ \underbrace{|w_\epsilon(s_i, \delta_{rr',j}) - w_\epsilon(S, \Delta_{rr'})|}_{3}] \tag{85}$$

Bounding (1) by Assumption 7:

$$|\hat{w}_\epsilon(s, \delta_{rr'}) - \hat{w}_\epsilon(s_i, \delta_{rr',j})| \tag{86}$$

$$= |\frac{1}{N(s, \delta_{rr'}, \epsilon_s, \epsilon_r)} \sum_{(k,k') \in N(s, \delta_{rr'}, \epsilon_s, \epsilon_r)} \frac{\prod_{t=1}^{H} \pi_e(a_t^k|s_t^k)\pi_e(a_t^{k'}|s_t^{k'})}{\prod_{t=1}^{H} \pi_b(a_t^k|s_t^k)\pi_b(a_t^{k'}|s_t^{k'})} \tag{87}$$

$$- \frac{1}{N(s_i, \delta_{rr',j}, \epsilon_s, \epsilon_r)} \sum_{(k,k') \in N(s_i, \delta_{rr',j}, \epsilon_s, \epsilon_r)} \frac{\prod_{t=1}^{H} \pi_e(a_t^k|s_t^k)\pi_e(a_t^{k'}|s_t^{k'})}{\prod_{t=1}^{H} \pi_b(a_t^k|s_t^k)\pi_b(a_t^{k'}|s_t^{k'})}| \tag{88}$$

$$\leq |\frac{1}{N(s, \delta_{rr'}, \epsilon_s, \epsilon_r)}(\sum_{(k,k') \in N(s, \delta_{rr'}, \epsilon_s, \epsilon_r)} \frac{\prod_{t=1}^{H} \pi_e(a_t^k|s_t^k)\pi_e(a_t^{k'}|s_t^{k'})}{\prod_{t=1}^{H} \pi_b(a_t^k|s_t^k)\pi_b(a_t^{k'}|s_t^{k'})} - \sum_{(k,k') \in N(s_i, \delta_{rr',j}, \epsilon_s, \epsilon_r)} \frac{\prod_{t=1}^{H} \pi_e(a_t^k|s_t^k)\pi_e(a_t^{k'}|s_t^{k'})}{\prod_{t=1}^{H} \pi_b(a_t^k|s_t^k)\pi_b(a_t^{k'}|s_t^{k'})})| \tag{89}$$

$$+ |(\frac{1}{N(s, \delta_{rr'}, \epsilon_s, \epsilon_r)} - \frac{1}{N(s_i, \delta_{rr',j}, \epsilon_s, \epsilon_r)}) \sum_{(k,k') \in N(s_i, \delta_{rr',j}, \epsilon_s, \epsilon_r)} \frac{\prod_{t=1}^{H} \pi_e(a_t^k|s_t^k)\pi_e(a_t^{k'}|s_t^{k'})}{\prod_{t=1}^{H} \pi_b(a_t^k|s_t^k)\pi_b(a_t^{k'}|s_t^{k'})}|] \tag{90}$$

$$\leq 2d^{2H} \frac{p_{\max}}{p_{\min}} \frac{\text{Vol}(\text{Diff}(B(s, \delta_{rr'}, \epsilon_s, \epsilon_r), B(s_i, \delta_{rr',j}, \epsilon_s, \epsilon_r)))}{Vol(B(\epsilon_s, \epsilon_r))} \tag{91}$$

$$= \tilde{\mathcal{O}}(\frac{2d^{2H} p_{\max} \epsilon_s^0 \epsilon_s^{d_s-1} \epsilon_r^0}{p_{\min} \epsilon_s^{d_s} \epsilon_r}), \tag{92}$$

where the last equation is followed by Li (2011).

Bounding (2) by Lemma 9:

Because with probability $\geq 1 - \delta$,

$$|\hat{w}_\epsilon(s_i, \delta_{rr',j}) - w_\epsilon(s_i, \delta_{rr',j})| \leq (C_{ips}^2 - c_{ips}^2)\sqrt{\frac{\ln(2N_S(\epsilon_s^0)N_{\Delta_{rr'}}(\epsilon_r^0)/\delta)}{N_{\min}}}, \forall i, j, \tag{93}$$

let $t = (C_{ips}^2 - c_{ips}^2)\sqrt{\frac{\ln(2N_S(\epsilon_s^0)N_{\Delta_{rr'}}(\epsilon_r^0)/\delta)}{N_{\min}}}$, we have $\delta = 2N_S(\epsilon_s^0)N_{\Delta_{rr'}}(\epsilon_r^0)e^{-(\frac{t}{C_{ips}^2 - c_{ips}^2})^2 N_{\min}}$, so

$$P^{\pi_b}(|\hat{w}_\epsilon(s_i, \delta_{rr',j}) - w_\epsilon(s_i, \delta_{rr',j})| \geq t) \leq 2N_S(\epsilon_s^0)N_{\Delta_{rr'}}(\epsilon_r^0)e^{-(\frac{t}{C_{ips}^2 - c_{ips}^2})^2 N_{\min}}.$$

$$\mathbb{E}^{\pi_b}[|\hat{w}_\epsilon(s_i, \delta_{rr',j}) - w_\epsilon(s_i, \delta_{rr',j})|] = \int_0^\infty P^{\pi_b}(|\hat{w}_\epsilon(s_i, \delta_{rr',j}) - w_\epsilon(s_i, \delta_{rr',j})| \geq t)dt \tag{94}$$

$$\leq \int_0^\infty 2N_S(\epsilon_s^0)N_{\Delta_{rr'}}(\epsilon_r^0)e^{-(\frac{t}{C_{ips}^2 - c_{ips}^2})^2 N_{\min}}dt \tag{95}$$

$$= \frac{(C_{ips}^2 - c_{ips}^2)N_S(\epsilon_s^0)N_{\Delta_{rr'}}(\epsilon_r^0)\sqrt{\pi}}{\sqrt{N_{\min}}} \tag{96}$$

$$= \tilde{\mathcal{O}}\Big(\frac{(1 + \frac{1}{\epsilon_s^0})^{d_s}(1 + \frac{1}{\epsilon_r^0})}{\sqrt{np_{\min}\epsilon_s^{d_s}\epsilon_r}}\Big) \tag{97}$$

Bounding (3) by Lipschitz property:

$$\mathbb{E}^{\pi_b}[|w_\epsilon(s_i, \delta_{rr',j}) - w_\epsilon(S, \Delta_{rr'})|] \leq L_s\epsilon_s + L_r\epsilon_r. \tag{98}$$

Putting it all together,

$$\mathbb{E}^{\pi_b}|\hat{w}_\epsilon(S, \Delta_{rr'}) - w(S, \Delta_{rr'})| = \tilde{\mathcal{O}}\Big(\frac{2d^{2H}p_{\max}\epsilon_s^0\epsilon_s^{d_s-1}\epsilon_r^0}{p_{\min}\epsilon_s^{d_s}\epsilon_r} + \frac{(1 + \frac{1}{\epsilon_s^0})^{d_s}(1 + \frac{1}{\epsilon_r^0})}{\sqrt{np_{\min}\epsilon_s^{d_s}\epsilon_r}} + \epsilon_s + \epsilon_r\Big) \tag{99}$$

$$= \tilde{\mathcal{O}}(n^{-1/2}\epsilon_s^{-3d_s/2}\epsilon_r^{-3/2} + \epsilon_s + \epsilon_r), \tag{100}$$

where the last step follows by setting $\epsilon_s^0 = \epsilon_s, \epsilon_r^0 = \epsilon_r$.

The rest of the proof follows directly from Proposition 2 in (Foffano et al., 2023). $\qquad\square$

### E.4 Proof of Theorem 2

We use Assumptions 4 to 6, which are standard in prior OPE literature.

*Proof.* Because

$$\mathbb{E}_{\pi_b}[\tilde{J}_{\text{IS}}(\tau_i)] = \mathbb{E}_{\pi_b}\Big[\frac{\prod_{t=1}^H \pi_e(a_t^i|s_t^i)}{\prod_{t=1}^H \pi_b(a_t^i|s_t^i)}J(\tau_i)\Big] = V^{\pi_e}, \tag{101}$$

and

$$\mathbb{E}_{\pi_b}[\tilde{J}_{\text{PDIS}}(\tau_i)] = \mathbb{E}^{\pi_b}\Big[\sum_{t=1}^H \gamma^{t-1}\prod_{k=1}^t \frac{\pi_e(a_k^i \mid s_k^i)}{\pi_b(a_k^i \mid s_k^i)}r_t\Big] = V^{\pi_e}, \tag{102}$$

the theorem with IS and PDIS is thus a direct consequence of Proposition 1 in (Angelopoulos et al., 2023). For WIS, by (Powell & Swann, 1966),

$$\mathbb{E}_{\pi_b}[\tilde{J}_{\text{WIS}}(\tau_i)] = \mathbb{E}^{\pi_b}\Big[n\frac{\prod_{t=1}^H \frac{\pi_e(a_t^i|s_t^i)}{\pi_b(a_t^i|s_t^i)}}{\sum_{i=1}^n \prod_{t=1}^H \frac{\pi_e(a_t^i|s_t^i)}{\pi_b(a_t^i|s_t^i)}}J(\tau_i)\Big] = V^{\pi_e} + O(\frac{1}{n}). \tag{103}$$

We still have

$$\tilde{J}_{\text{WIS}}(\tau_i) - V^{\pi_e} = \tilde{J}_{\text{WIS}}(\tau_i) - \mathbb{E}_{\pi_b}[\tilde{J}_{\text{WIS}}(\tau_i)] + \mathbb{E}_{\pi_b}[\tilde{J}_{\text{WIS}}(\tau_i)] - V^{\pi_e} = O_p(\frac{1}{\sqrt{n}}) + O(\frac{1}{n}) = O_p(\frac{1}{\sqrt{n}}), \tag{104}$$

which indicates the desired result following the standard proof of Proposition 1 in (Angelopoulos et al., 2023).

$\qquad\square$

### E.5 Variance of `DR-PPI`

The estimator for the first fold is

$$\widehat{V}_{\text{DR-PPI:1}}^{\pi_e} = \underbrace{\frac{1}{N_f} \sum_{i=1}^{N_f} J(\tilde{\tau}_i)}_{A} + \underbrace{\frac{1}{n/2} \sum_{j \in D_2} \tilde{J}(\tau_j)}_{B} - \underbrace{\frac{1}{n/2} \sum_{j \in D_2} \frac{1}{M} \sum_{m=1}^{M} J(\tilde{\tau}_{m,j} \mid s_{0,j})}_{C}. \tag{105}$$

Throughout, we condition on the generative model $f_1$ learned from $D_1$, treating it as fixed. We use the following notation:

$$\sigma_{f_1}^2 := \text{Var}\big(J(\tilde{\tau})\big), \qquad \text{marginal variance of synthetic returns (e.g., averaged across all initial states),}$$

$$\sigma_{f_1}^2(s_0) := \text{Var}\big(J(\tilde{\tau}) \mid s_0\big), \quad \text{conditional variance of synthetic returns,}$$

$$\sigma_r^2(s_0) := \text{Var}\big(\tilde{J}(\tau) \mid s_0\big), \quad \text{conditional variance of IS-weighted real returns,}$$

$$\mu_{f_1}(s_0) := \mathbb{E}\big[J(\tilde{\tau}) \mid s_0\big], \qquad \text{conditional mean synthetic return (expectation taken over trajectories from } f_1\text{),}$$

$$\mu_r(s_0) := \mathbb{E}\big[\tilde{J}(\tau) \mid s_0\big], \qquad \text{conditional mean IS-weighted return.}$$

**Proposition 10** (Variance of the single-fold DR-PPI estimator)**.** *Under the independence structure described below,*

$$\text{Var}\big(\widehat{V}_{\text{DR-PPI:1}}^{\pi_e}\big) = \frac{\sigma_{f_1}^2}{N_f} + \frac{1}{n/2}\Big[\mathbb{E}_{s_0}\big[\sigma_r^2(s_0)\big] + \frac{\mathbb{E}_{s_0}\big[\sigma_{f_1}^2(s_0)\big]}{M} + \text{Var}_{s_0}\big(\mu_r(s_0) - \mu_{f_1}(s_0)\big)\Big]. \tag{106}$$

*Proof.* Term $A$ consists of synthetic trajectories $\tilde{\tau}_i$ drawn i.i.d. from $f_1$, independently of the real dataset $D_2$. Terms $B$ and $C$ both depend on $D_2$: $B$ through the IS-weighted returns $\tilde{J}(\tau_j)$, and $C$ through the initial states $s_{0,j} \subset \tau_j$. Since $A$ is constructed entirely from synthetic trajectories drawn from $f_1$ that do not depend on $D_2$, we have

$$A \perp (B, C).$$

Therefore $\text{Cov}(A, B - C) = 0$, and

$$\text{Var}\big(\widehat{V}_{\text{DR-PPI:1}}^{\pi_e}\big) = \text{Var}(A) + \text{Var}(B - C). \tag{107}$$

The trajectories $\tilde{\tau}_1, \ldots, \tilde{\tau}_{N_f}$ are i.i.d. draws from $f_1$, so by standard variance of an empirical mean,

$$\text{Var}(A) = \frac{\sigma_{f_1}^2}{N_f}. \tag{108}$$

Note that this term can approach 0 if $N_f$ is extremely large.

Define the per-trajectory residual

$$Z_j := \tilde{J}(\tau_j) - \frac{1}{M} \sum_{m=1}^{M} J(\tilde{\tau}_{m,j} \mid s_{0,j}), \qquad j \in D_2,$$

so that $B - C = \frac{1}{n/2} \sum_{j \in D_2} Z_j$. Because the real trajectories $\tau_j$ are i.i.d. and the synthetic rollouts $\tilde{\tau}_{m,j}$ are drawn independently for each $j$, the $Z_j$ are mutually independent and identically distributed. Hence

$$\text{Var}(B - C) = \frac{\text{Var}(Z_j)}{n/2}. \tag{109}$$

Condition on the initial state $s_{0,j}$:

$$\text{Var}(Z_j) = \underbrace{\mathbb{E}_{s_0}\big[\text{Var}(Z_j \mid s_0)\big]}_{\text{within-state noise}} + \underbrace{\text{Var}_{s_0}\big(\mathbb{E}[Z_j \mid s_0]\big)}_{\text{across-state noise}}. \tag{110}$$

Conditional on $s_0$, the real trajectory $\tau_j$ and the synthetic trajectory $\{\tilde{\tau}_{m,j}\}$ are drawn from independent distributions, so $\tilde{J}(\tau_j)$ and $\frac{1}{M} \sum_m J(\tilde{\tau}_{m,j} \mid s_0)$ are conditionally independent. Thus

$$\text{Var}(Z_j \mid s_0) = \text{Var}\big(\tilde{J}(\tau_j) \mid s_0\big) + \text{Var}\left(\frac{1}{M} \sum_{m=1}^{M} J(\tilde{\tau}_{m,j} \mid s_0) \,\middle|\, s_0\right).$$

The first term is $\sigma_r^2(s_0)$ by definition. For the second, since the $M$ synthetic rollouts are i.i.d. given $s_0$,

$$\text{Var}\left(\frac{1}{M} \sum_{m=1}^{M} J(\tilde{\tau}_{m,j} \mid s_0) \,\middle|\, s_0\right) = \frac{\sigma_{f_1}^2(s_0)}{M}.$$

Taking the expectation over $s_0$:

$$\mathbb{E}_{s_0}\big[\text{Var}(Z_j \mid s_0)\big] = \mathbb{E}_{s_0}\big[\sigma_r^2(s_0)\big] + \frac{\mathbb{E}_{s_0}\big[\sigma_{f_1}^2(s_0)\big]}{M}. \tag{111}$$

The conditional mean of $Z_j$ given $s_0$ is

$$\mathbb{E}[Z_j \mid s_0] = \mathbb{E}\big[\tilde{J}(\tau_j) \mid s_0\big] - \mathbb{E}\left[\frac{1}{M} \sum_{m=1}^{M} J(\tilde{\tau}_{m,j} \mid s_0) \,\middle|\, s_0\right] = \mu_r(s_0) - \mu_{f_1}(s_0).$$

Therefore

$$\text{Var}_{s_0}\big(\mathbb{E}[Z_j \mid s_0]\big) = \text{Var}_{s_0}\big(\mu_r(s_0) - \mu_{f_1}(s_0)\big). \tag{112}$$

Substituting Equation (111) and Equation (112) into Equation (110):

$$\text{Var}(Z_j) = \mathbb{E}_{s_0}\big[\sigma_r^2(s_0)\big] + \frac{\mathbb{E}_{s_0}\big[\sigma_{f_1}^2(s_0)\big]}{M} + \text{Var}_{s_0}\big(\mu_r(s_0) - \mu_{f_1}(s_0)\big). \tag{113}$$

Substituting Equation (108), Equation (109), and Equation (113) into Equation (107) yields Equation (106).
$\qquad\square$

**Corollary 11** (Variance of the cross-fitted DR-PPI estimator). *Let $\widehat{V}_{\text{DR-PPI}}^{\pi_e} = \frac{1}{2}\big(\widehat{V}_{\text{DR-PPI:1}}^{\pi_e} + \widehat{V}_{\text{DR-PPI:2}}^{\pi_e}\big)$. Since the two folds use disjoint real data $D_1, D_2$ and independently trained generative models, the two fold estimators are approximately independent. By symmetry of the split, both folds have equal variance, so*

$$\text{Var}\big(\widehat{V}_{\text{DR-PPI}}^{\pi_e}\big) \approx \frac{1}{2}\text{Var}\big(\widehat{V}_{\text{DR-PPI:1}}^{\pi_e}\big) = \frac{\sigma_{f_1}^2}{2N_f} + \frac{1}{n}\left[\mathbb{E}_{s_0}\big[\sigma_r^2(s_0)\big] + \frac{\mathbb{E}_{s_0}\big[\sigma_{f_1}^2(s_0)\big]}{M} + \text{Var}_{s_0}\big(\mu_r(s_0) - \mu_{f_1}(s_0)\big)\right]. \tag{114}$$

