# OpenReview forum: "PERRY: Policy Evaluation with Confidence Intervals using Auxiliary Data"
_TMLR — Accepted by TMLR_

### Review · Reviewer_Jn7m · 2026-03-30

**Summary Of Contributions:**

The paper studies off-policy evaluation (OPE) problem with uncertainty quantification when both real offline trajectories and synthetic (model-generated) trajectories are available (i.e., with data augmentation). The main focus of the paper is on designing algorithms and guarantees such that the confidence intervals remain valid despite model bias in the augmented data. Specifically, the paper proposed two separate methods for two different inference targets: **CP-Gen** for state-conditioned values $V^{\pi_e}(s)$, which builds a conformal interval around the discrepancy between real and generated returns; and **DR-PPI** for the more general value function $V^{\pi_e}(d_0)$, which combines model-based target-policy rollouts with an importance-weighted correction term in a doubly-robust style. Accordingly, the paper provides theoretical result that validates the confidence interval choice in both algorithm; and somewhat comprehensive empirical studies across a variety of simulator domains and a real MIMIC-IV healthcare setting.

---

**Strengthes:**

1. **Interesting problem setting.** The paper tackles a genuinely important gap in OPE: once synthetic / model-generated trajectoreis are introduced, uncertainty quantification is no longer straightforward since both distribution shift and model misspecification incur. Hence this would lead to good motivation.

2. **Algorithmic innovation tailored to two different targets.** The paper does not force a single method onto two distinct tasks, but instead proposed **CP-Gen** and **DR-PPI** for fixed-initial-state and random-initial-state value, respectively. CP-Gen is technically novel in combining discrepency-based conformal prediction with local density-ratio weighting, while DR-PPI gives a natural model-based construction (with correction) for augmented OPE.

3. **Empirical study is fairly broad and supports the main qualitiative message.** The experiments span multiple domains including inventory control, sepsis, D4RL, HalfCheetah, and MIMIC-IV, and they consistently illustrate the paper's main empirical claim: naively adding synthetic data to existing OPE estimators can give overly optimistic, non-covering intervals, whereas the proposed methods more consistently preserve converage while sometimes yielding shorter valid intervals.

---

**Weaknesses:**

1. **The paper feels more like two related methods under one theme than a unifed framework (?).** CP-Gen and DR-PPI differ substantially in target parameter, technical machinery (e.g., one needs generation from behavior policy but another not), and theoretical flavor (e.g., one gives a non-asymptotic rate but another only asymptotic control). So the paper can feel heterogeneous.

2. **The DR-PPI guarantee is only asymptotic, with limited theoretical insight into efficiency or variance.** Theorem 2 gives asymptotic coverage, but there is no sharp variance analysis (as standard in many seminal OPE works, such as the DR paper[Jiang and Li, 2016]), or efficiency comparison of the type often emphasized in these seminal OPE works.

3. **The theory does not quantify how synthetic sample size improves performance (or does it?).** A central promise of augmentation is that more synthetic trajectories would help, but the theory (especially for CP-Gen) does not make the dependency on the number of generated rollouts particularly explicit (i.e., $M$ or $N$). So it remains unclear when, and by how much rate, increasing synthetic data should tighten intervals.

4. **Presentation issue.** Despite the notation table, the presentation in Section 3.1 and the deviration of CP-Gen are difficult to follow (at least for me. I take really long time to catch the essence of CP-Gen). For instance, the indexing switches between $i$ and $(j,n)$, quantities such as $M,N,n,K$ are easy to lose track of, and overloaded notation such as $V$ for both values and scores can make the section much harder than necessary to read (though the algorithm itself is much more "elegant").

**Audience:**

Yes

**Audience Explanation:**

I believe this paper would be of interest to at least a meaningful subset of the TMLR audience, especially researches working on offlien RL, OPE, uncertainty quantification, model-based RL, distribution (covariate) shift, and reliable decision-making in high-stakes domains. To be honest, the part that appeals me most is the motivation: standard OPE methods rely on reweighting (e.g., IS, WIS) or value estimation (e.g., DM, FQE); and many recent works explore using synthetic or model-generated data to improve offline evaluation, but much less is known about how to maintain valid confidence intervals in that setting. That problem is important both practically and methodologically, as claimed by authors in Section 1.

**Broader Impact Concerns:**

I have no additional broader-impact concrns beyond those already acknoledged by the authors; the existing Broader Impact Statement is sufficient for this submission.

**Claims And Evidence:**

Yes

**Claims Explanation:**

The paper's main qualitative claims are reasonably supported since it clearly demonstrates that naively incorporating synthetic trajectories into OPE can lead to overly wide CIs that fail to cover, and both the theory and experiments support the claim that the propsed methods can produce more reliable intervals in such settings. The empirical study is broad enough enough to back up this central message, and the theoretical sections do provide formal guarantees for both method, i.e., an approximate coverage guarantee for CP-Gen and an asymptotic coverage guarantee for DR-PPI. In that sense, the core claims are supported.

That said, I would not say all claims are supported equally strongly. In particular, the theoretical results do not very directly quantify when augmentation helps as a function of the amount or quality of synthetic data, and for DR-PPI the CI guarantee is only asymptotic (see the weaknesses above). Also a subtle point: the "PERRY" phrase only appears in the title, but not in the contents.

**Requested Changes:**

1. Clarify the relationship between the proposed methods and standard OPE baselines (especially DR/IS), not empirically. In particular, please explain more explicitly in what sense DR-PPI is "doubly-robust" (since current presentation only shows one side of the robustness, i.e., the asymptotic result) and how it differs from classical DR. And, if possible, provide a variance analysis of DR-PPI.

2. Improve the presentation of Section 3.1 / Algorithm 1. Add more implementation details for Algorithm 1 (e.g., the generative model $\mathcal{T}$). Some notational problems are mentioned in weaknesses.

3. Better explain why the paper needs two rather different algorithms. And discuss on why random-initial-state would change the problem so much, since many works on offline RL theory just treat the initial state $s_0$ as fixed (so as the initial action $a_0$) and then treat the first transition as a random state distribution, in the finite-horizon MDP setting.

4. Better discuss on the synthetic augmentation procedure and when such data augmentation helps.

---

> ### Author Response · Authors · 2026-04-16
>
> We thank the reviewer for their comments, and are thrilled to hear that our work “tackles a genuinely important gap in OPE”. We respond to individual concerns below.
>
> **Necessity of two OPE estimators**: The two proposed methods should be viewed as complementary solutions to the uncertainty-aware OPE problem, tailored to two different evaluation targets. The methods differ in two ways: (1) the strategy used to provide valid uncertainty estimates, and (2) the evaluation target.
>
> CP-Gen targets the initial-state-conditioned value $V^{\pi_e}(s)$ and uses weighted conformal prediction to obtain finite-sample coverage for the return discrepancy at a fixed state. DR-PPI instead targets the population-average policy value $V^{\pi_e}$ and uses a DR / PPI correction to obtain an asymptotically valid confidence interval. In principle, DR-PPI can also be applied to estimate initial-state-conditioned values, and CP-Gen could be aggregated across initial states. However, these approaches would lead to different trade-offs in terms of statistical guarantees and interval tightness.
>
> We agree that many works in offline RL condition on a fixed $s_0$ for simplicity. However, in our setting, the distinction between conditional and marginal evaluation becomes important because uncertainty quantification behaves differently under these two targets. In particular, constructing valid confidence intervals for $V^{\pi_e}(s)$ requires controlling the distribution of trajectory returns conditioned on a specific initial state, whereas estimating $V^{\pi_e}$ involves averaging over the initial state distribution $d_0$, which allows for different variance reduction strategies (e.g., via aggregation and cross-fitting).
>
> **Efficiency of DR-PPI**: Our primary theoretical goal for DR-PPI is to establish valid confidence intervals under the use of synthetic data, and Theorem 2 focuses on asymptotic coverage. The estimator is closely connected to the prediction-powered inference (PPI) framework, and PPI estimators are known to achieve variance reduction and can attain asymptotic efficiency under mild conditions [2].
>
> In addition, in our work we derive an explicit variance expression for the DR-PPI estimator (Equation (16) in the updated manuscript). This decomposition provides insight into when DR-PPI is expected to be effective. When the generative model is accurate and the environment is sufficiently stochastic, the model-based component reduces variance, leading to tighter confidence intervals compared to purely IS-based methods. We also include a proof of the variance of DR-PPI in Appendix E.5.
>
> **How many synthetic trajectories to generate?**:
> For CP-Gen, our theoretical analysis establishes validity of the conformal interval under estimated weights $\hat{w}_{\epsilon}$.
>
> Rather than explicitly parameterizing the number of synthetic trajectories, we assume it is large enough that $\hat{w}_{\epsilon}$
>
> satisfies the required moment and concentration conditions (e.g., $\mathbb{E}{\pi_b}[|\hat{w}\epsilon(S,\Delta{rr'})|^k] \leq d^{2k}$). We express the coverage error in terms of weight estimation quality rather than directly in terms of $M$. Making this dependence explicit would require a more detailed analysis of how $\hat{w}_\epsilon$ concentrates as a function of synthetic rollouts, which we leave for future work.
> For DR-PPI, the theoretical guarantee similarly considers the regime where $M \to \infty$, ensuring the model-based component is well-estimated. The resulting confidence interval is then driven by the variance of the combined estimator.
>
> In practice, generating synthetic trajectories is cheap once a model is trained, and more rollouts reduce the variance of the model-based component. We also empirically study the ffect of the number and quality of generated trajectories in (Figure 1).
>
> **Notation and Presentation**:
> In the revised manuscript, we have improved the presentation of Section 3.1 by standardizing notation and simplifying the presentation.
>
> **Relationship between DR-PPI and OPE baselines**:
> DR-PPI can be viewed as a hybrid between classical DR estimators and prediction-powered inference (PPI). Similar to DR estimators, DR-PPI combines a model-based estimate with a correction term. Unlike standard DR estimators, which rely on a learned value function, DR-PPI uses Monte Carlo rollouts from a generative model. This leads to a different variance structure and allows us to incorporate large amounts of synthetic data. We have included a variance proof of DR-PPI in the appendix [TODO: Insert section].
>
> **Synthetic augmentation procedure**:
> Synthetic trajectories are generated by rolling out a learned dynamics model under a specified policy. We have added further details on the synthetic augmentation approach in Appendix B.
>
> [1] Gao et al. "Variational latent branching model for off-policy evaluation." 2023.
> [2] Ji et. al. "Predictions as surrogates: Revisiting surrogate outcomes in the age of ai."(2025).

---

### Review · Reviewer_vfdH · 2026-04-03

**Summary Of Contributions:**

The paper addresses the problem of quantifying uncertainty when using data augmentation for Off-Policy Evaluation, by leveraging multiple techniques, namely: Conformal Prediction, Doubly Robust estimation and Prediction Powered Inference. The paper proposes two methods estimating a confidence interval over the state-value function (CP-gen) and value (DR-PPI) of a given target policy. The methods are formally and empirically validated by the authors.

Strengths:

- The developed methods are intuitive and easy to implement
- Both techniques come with some theoretical guarantees
- The evaluation is comprehensive

Weaknesses (see the requested changes for more details):
- The writing is sometimes confusing, especially in the Problem Setting and Method sections
- Some of the modeling choices are not clearly justified

**Audience:**

Yes

**Audience Explanation:**

I believe uncertainty quantification is a subject of great interest among a wide range of researchers, from both the theoretical and practical point of view.

**Claims And Evidence:**

Yes

**Claims Explanation:**

The theoretical and empirical results support the claims of the authors, showing that the proposed methods can learn correct confidence intervals with good coverage.

**Requested Changes:**

1. (**critical**) I believe in the problem setting it would be useful to define more formally the core techniques used in the paper, that not all readers might be familiar with. That would be Conformal Prediction (classical and weighted), Doubly Robust estimation and Prediction Powered Inference. While they are discussed in the related work section, it would still be good to formally define what these methods aim to achieve (and how), and why they are helpful in the context of OPE.

2. (**critical**) The step from Eq. 3 to Eq. 4 should be defined more precisely: in Eq. 3 we have an expectation over $p^{\pi_e}$ that in Eq. 4 is over data from the behavior policy $\pi_b$. I imagine there should be a reweighting term somewhere, which is discussed later on in the section. I believe it would be nice to introduce the distribution shift problem around Eq. 4 (and maybe even in the problem definition section, since it is a core challenge of OPE), and then proceed to explain the rest of the section.

3. In Algorithm 2, lines 7 and 8 use equations 23 and 24, which are defined in the Appendix. I think for readability and understanding purposes it might be useful to present those equations before Algorithm 2 (or in the main body of the paper anyway).

4. In Eq. 14, $\tilde \tau_{m,j}$ is defined after one paragraph. I think it would be nice to define it either before or immediately after the equation.

5. While it is true that the theory is model-agnostic, I would have appreciated a more clear definition of what you consider as a "model" to generate trajectories. Are you considering an autoregressive model or a "trajectory segment" generative model (see for example [1],[2] and other works in the generative model-based area). This does not impact the theoretical soundness but can help to understand how the trajectories are generated under a policy different from the behavior one.

6. The empirical results are extensive and well documented. Have you considered conducting ablation studies on the choice of the coverage level $\alpha$? I feel like in many Conformal Prediction papers the default choice is $\alpha = 0.1$, while you choose $\alpha = 0.05$. By choosing a low alpha, we get a higher probability that the sample will be contained in the confidence interval but at the same time there is a risk that the confidence interval will be trivial (i.e., very large).

5. Towards the end of section 5, you say that DR-PPI underperforms if the model is of poor quality, for example in the HalfCheetah and MIMIC-IV environments. Why is the quality of the model poor in those environments? I think the claim should be justified in the paper.

6. (**critical**) Why does table 4 present the coverage rates only for Inventory and Sepsis? Maybe you can include the other two environments in the Appendix (right now I think you only show the interval size)?

7. It would be nice from a reproducibility point of you if you could include some hyperparameters for the learned models (or the code, at the end of the review process to avoid violating anonymity).

8. (**critical**) I cannot find a proof for Lemma 9 in the Appendix.

---

> ### Comment · Reviewer_vfdH · 2026-04-03
> **Missing references**
>
> Sorry, I just realized I forgot to insert the two references for point 5. Here they are:
>
> [1] Janner, Michael, et al. "Planning with diffusion for flexible behavior synthesis." arXiv preprint arXiv:2205.09991 (2022).
>
> [2] Rigter, Marc, Jun Yamada, and Ingmar Posner. "World models via policy-guided trajectory diffusion." arXiv preprint arXiv:2312.08533 (2023).

---

> > ### Author Response · Authors · 2026-04-16
> >
> > We thank the reviewer for their feedback and we are grateful to hear that our work contributes to “a subject of great interest”. We respond to individual concerns below.
> >
> > **Formal definition of core techniques**:
> > In the revised manuscript, we have (i) introduced the standard conformal prediction framework and its coverage guarantee, (ii) described weighted conformal prediction and how importance weights correct for distribution shift, and (iii) provided background on PPI.
> >
> > **Step from Eq. 3 to Eq. 4**:
> > We would like to clarify that the derivation from Eq. (3) to Eq. (4) does not involve reweighting. In Eq. (3), both $\tau$ and $\tilde{\tau}$ are from the target policy $p^{\pi_e}$, and Eq. (4) approximates the expectation in the “model bias/return discrepancy” term using empirical averages.
> >
> > The key challenge arises because we only observe trajectories from the behavior policy $\pi_b$. As a result, the empirical return differences $J(\tau \mid s_0 = s) - J(\tilde{\tau} \mid s_0 = s)$ available for calibration are drawn from a distribution induced by $\pi_b$, rather than $\pi_e$.  This mismatch introduces a distribution shift between the target quantity of interest (return differences under $\pi_e$) and the available data (pairs constructed under $\pi_b$). Weighted conformal prediction is then used to correct for this shift by reweighting samples according to likelihood ratios between $\pi_e$ and $\pi_b$, which we formalize in Eq. (8). We will revise the manuscript to clarify this.
> >
> > **Restructuring Notation**:
> > We have moved equations 23/24 to the main text and defined $\tilde{\tau}$ before Equation 14.
> >
> > **Defining the generative model**:
> > In our framework, the key requirement for the generative model is the ability to conditionally generate trajectories under a specified policy. Concretely, given a state $s_t$ and an action $a_t \sim \pi(\cdot \mid s_t)$ , the model must be able to generate the next state and reward.
> >
> > In our experiments, we consider both a VAE and a neural network to approximate the transition dynamics. However, other classes of generative models can also be used. For example, diffusion-based models can be adapted to generate trajectories conditioned on actions sampled from $\pi_e$.
> >
> > We have revised the manuscript (Section 2.5) to explicitly state this requirement and provide examples of compatible model classes.
> >
> > **Choosing $\alpha$**:
> > We agree that the choice of the coverage level $\alpha$ plays an important role in determining the trade-off between confidence and interval width. In our experiments, we fix $\alpha = 0.05$ across all methods. This is standard practice in the OPE and uncertainty quantification literature, where methods are typically compared at a fixed confidence level (e.g., 95% confidence intervals), and performance is evaluated based on coverage and interval width[1][2][3].
> >
> > While varying $\alpha$ can provide additional insight into this trade-off, we note that the qualitative behavior is well understood: smaller $\alpha$ yields higher coverage with wider intervals, and vice versa. Our primary goal is to demonstrate that our methods achieve valid coverage while producing tighter intervals than baselines at a standard confidence level.
> >
> > We agree that choosing $\alpha$ should be informed by domain-specific considerations, particularly in high-stakes settings where statistical confidence and decision utility must be balanced. We believe that a systematic study of this trade-off is beyond our scope.
> >
> > **DR-PPI Performance**:
> >
> > We thank the reviewer for catching this and apologize for the confusion. The claim about poor model quality in HalfCheetah was a mistake; our VAE dynamics model approximates the true dynamics reasonably well there. The real limiting factor is the environment's near-deterministic nature, which reduces the variance reduction benefit of synthetic rollouts beyond standard importance sampling.
> >
> > For MIMIC-IV, the VAE model quality suffers due to the small dataset (~400 trajectories) and complex dynamics (state dim = 20, avg. horizon ~30). This leads to biased synthetic trajectories, which weakens the performance of DR-PPI. We have revised the manuscript to reflect these points.
> >
> > **Coverage rates for D4RL and MIMIC**:
> > We have updated Table 4 to include coverage rates for the D4RL and MIMIC settings.
> >
> > **Hyperparameters and code**:
> > We will release the full implementation upon completion of the review process. We have anonymized code at https://anonymous.4open.science/r/perry_anon-D762/README.md, which includes all details to reproduce our results.
> >
> > **Lemma 9 proof**:
> > We have included the proof of Lemma 9 in Appendix E.2.
> >
> > [1] Angelopoulos, Anastasios N., et al. "Prediction-powered inference." (2023)
> >
> > [2] Hanna, J., Stone, P., & Niekum, S. (2017, February). Bootstrapping with models: Confidence intervals for off-policy evaluation.
> >
> > [3] Thomas, Philip, Georgios Theocharous, and Mohammad Ghavamzadeh. "High-confidence off-policy evaluation." (2015)

---

### Review · Reviewer_3XzM · 2026-04-07

**Summary Of Contributions:**

## Summary

The paper proposes two methods to compute confidence intervals for the setting where synthetic data is used alongside offline data. These methods are:

1) CP-Gen: Compute confidence intervals for the state-conditioned value function under the evaluation policy using conformal prediction, and further accounting for the mismatch between behavioral and evaluation policy. In order to scale their approach to continuous high-dimensional state spaces, they used $\epsilon$ approximation while computing the Importance weights.

2) DR-PPI: Compute confidence intervals for the return of the evaluation policy, using a Doubly-Robust (DR) method

The paper produces theoretical guarantees that its proposed approaches produce valid confidence intervals. Experimental results show that their approaches produce valid confidence intervals across 4 domains.

## Strengths

1) The setting is novel. The setting of providing rigorous uncertainty estimates for OPE in MDPs while using synthetic data augmentation is relatively unexplored.

2) The paper produces theoretical guarantees for its two proposed approaches. Finite sample guarantees for CP-Gen and asymptotic guarantees for DR-PPI.

## Weaknesses

1. The paper uses $\epsilon$ balls around states and return differences in order to make their proposed CP-Gen approach work with continuous high-dimensional state spaces. But in high dimensions (> 100), their proposed approach would be affected by the curse of dimensionality. For a fixed $\epsilon$ as the dimension increases, the volume of the ball reduces, leading to a very small number of data points available to compute the IS ratio, leading to highly noisy estimates. On the other hand, as the dimension increases, maintaining a fixed number of points inside the ball requires $\epsilon$ to grow, hurting the coverage as per Theorem 1. This casts doubt on whether their approach can truly handle continuous high-dimensional state spaces.

2. To add to the above point, the domains considered in the experiments have at most a state dimension of 20. To truly demonstrate that their approach handles high-dimensional states, the authors should include other domains with higher dimensions (> 100) (e.g., Atari). Furthermore, Table 4 demonstrates a drop in coverage even for a dimension of 8 (Sepsis), casting doubt on whether their proposed approach truly handles high-dimensional state spaces.

3. Theory-Practice mismatch: The authors mention in their practical considerations that they clip importance weights to address blow-up. It is unclear if this is considered in their theorems demonstrating that their proposed approaches produce valid confidence intervals.

4. Equation 15 in DR-PPI uses the normal distribution CI. Importance weighted return distributions have high variance and flatter tails. It is unclear why something like a Student-$t$ CI bound was not used here and if the normal distribution CI is even valid. The authors should use CIs that consider flatter-tailed distributions in the DR-PPI approach.

5. The authors mention that DR-PPI produces asymptotically valid CIs in Section 4.2. No theory is provided on how this validity would hold in the finite-sample setting, and which correction (IS, WIS, PDIS) fares better in this setting.

6. Related work is not fleshed out properly. The authors should provide a small segment about conformal prediction for readers uninformed about it. Additionally, Equation 11 shows the expression for the conformal band. How the return differences that satisfy the constraint are determined is missing from the background.

7. $\alpha$ is treated as a confidence parameter in Equation 6, but as an error rate everywhere else. The authors should be consistent with the notation for what $\alpha$ represents.

8. The proposed CP-Gen approach depends on $\epsilon_{s}$ and $\epsilon_{r}$. It is unclear how to choose these values effectively and how these values need to be changed as the dimensionality of the state space increases.

### Minor Nits

1) What does the title PERRY have to do with the paper?

**Audience:**

Yes

**Audience Explanation:**

The paper addresses the problem of providing valid confidence intervals for OPE when using synthetically generated data alongside offline data, a setting that has been relatively unexplored in the literature. This is an important and timely contribution, particularly for high-stakes domains such as healthcare, where reliable uncertainty estimates are critical for safe policy deployment.

**Claims And Evidence:**

No

**Claims Explanation:**

The paper repeatedly claims to handle high-dimensional continuous state spaces , in the abstract, related work, and empirical sections, yet the experiments consider a maximum state space dimension of 20. This is a significant mismatch between claims and evidence. Furthermore, Table 4 already shows a 3 percentage point drop in coverage at $d_s = 8$ (Sepsis), relative to the requested level. The authors do not study how coverage deteriorates as $d_s$ increases, nor do they provide guidance on how $\epsilon_s$ should be chosen as dimensionality grows. The authors should either include experiments on genuinely high-dimensional domains ($d_{s} > 100$, e.g., Atari) to substantiate their claims, or tone down their comments about handling high-dimensional state spaces.

**Requested Changes:**

## Proposed Adjustments

### Critical

1. **High-dimensional state spaces:** The paper repeatedly claims to handle high-dimensional continuous state spaces, in the abstract, related work, and empirical sections, yet the experiments consider a maximum state space dimension of 20. Table 4 already shows a 3 percentage point drop in coverage at $d_s = 8$ (Sepsis), and coverage results are not reported for the two highest-dimensional domains (HalfCheetah, MIMIC-IV). The authors should either include experiments on genuinely high-dimensional domains ($d_s > 100$, e.g., Atari) to substantiate their claims, or tone down their comments about handling high-dimensional state spaces. In the current version, this claim is not supported by the evidence provided. Additionally, the authors should emphasize how coverage deteriorates as $d_s$ increases and provide guidance on how $\epsilon_s$ should be chosen as dimensionality grows.

### Would Strengthen the Work

2. **Notation consistency:** $\alpha$ is treated as a confidence parameter in Equation 6 but as an error rate everywhere else. The authors should correct Equation 6 to read $\geq 1 - \alpha$ and ensure consistent use throughout.

3. **Theory-practice mismatch on clipping:** The authors clip importance weights in practice but it is unclear whether this is accounted for in Theorems 1 and 2. The authors should clarify whether their theoretical guarantees hold under clipping.

4. **Normal approximation in DR-PPI:** Equation 15 uses a standard normal CI despite importance-weighted return distributions being known to have heavy tails. The authors should either justify the normal approximation empirically or discuss heavier-tailed alternatives such as Student-$t$.

5. **Finite-sample behavior of DR-PPI:** Theorem 2 only provides asymptotic validity. The authors should provide guidance on finite-sample behavior and which IS correction (IS, WIS, PDIS) is preferable in practice.

6. **Background on conformal prediction:** The related work section does not provide sufficient background on conformal prediction for readers unfamiliar with it. A brief self-contained summary would improve accessibility.

---

> ### Author Response · Authors · 2026-04-08
> **Quick Clarification**
>
> We thank the reviewer for their thorough and constructive feedback.
>
> Before we prepare our full response, we wanted to get clarification on one point. The review mentions two possible paths forward: (1) including experiments on high-dimensional domains (e.g., Atari), or (2) toning down the claims about high-dimensional state spaces throughout the paper. Would adjusting the language of our claims be sufficient to address this concern?
>
> Thank you again for your time and we look forward to your clarification.

---

> > ### Comment · Reviewer_3XzM · 2026-04-09
> >
> > I would be ok with adjusting the language about performance on high-dimensional state spaces. If you choose to do this, please also provide results for the other 2 domains in Table 4 (D4RL-Half Cheetah and MIMIC-IV). Additionally, please add a small paragraph highlighting potential issues when scaling to high dimensions, and guidance on choosing $\epsilon$ for the IS ratio approximation in CP-Gen

---

> > > ### Author Response · Authors · 2026-04-16
> > >
> > > We thank the reviewer for their thorough feedback. We respond to individual concerns below.
> > >
> > > **High dimensional spaces**:
> > > We agree that the language overstates the scope of our empirical validation, and we have revised the manuscript accordingly. Our revised framing situates our contribution relative to prior work (Foffano et. al), which is restricted to tabular settings, and characterizes our method as extending tractability to low-to-moderate-dimensional continuous domains.
> > >
> > > We have also added coverage results for HalfCheetah and MIMIC-IV to the updated manuscript (Table 4).
> > >
> > > **Notation Consistency**:
> > > We apologize for the confusion with this notation. Equation 6 should contain $1-\alpha$ where $\alpha$ is the confidence level, and we have updated the manuscript to reflect this.
> > >
> > >
> > > **Clipping Importance Weights**:
> > > We acknowledge that clipping importance weights introduces an additional bias that is not accounted for in Theorems 1 and 2, and we have added a clarifying remark to this effect in the manuscript. However, we only use weight clipping in our HalfCheetah experiments, where it is a practical necessity due to the variance of importance weights in continuous action spaces as noted by prior work [1].
> > >
> > > Our theoretical results in Theorems 1 and 2 apply directly and without modification to all other experimental settings, where no clipping is performed.
> > >
> > > **Normal approximation for DR-PPI**:
> > > We argue that the distribution of state values, which is what Equation 15 operates over, is unlikely to be heavy-tailed. This is because the value estimates are bounded by the range of cumulative rewards, and averaging across the state distribution further suppresses tail behavior. The normal approximation for value-based quantities is standard practice in the offline RL and off-policy evaluation literature. It is employed in doubly robust estimators [2] and also in prediction-powered inference [3]. We will add a brief justification of this approximation to the manuscript, and note that heavier-tailed alternatives such as the Student-t could be adopted as a drop-in replacement if warranted by a particular application, though we expect the practical difference to be negligible in the regimes we consider.
> > >
> > > **Finite sample behavior of DR-PPI**:
> > > We direct the reviewer to our empirical results, which directly characterize finite-sample behavior across a range of datasets and demonstrate that the coverage guarantees hold reliably even in moderate-sample regimes. Beyond empirical evidence, we also provide a closed-form variance calculation for our estimator (Equation 16, proof in Appendix E.5), which can be used to analytically characterize finite-sample uncertainty and inform practitioners on the sample sizes required for reliable coverage.
> > >
> > > Regarding the choice of IS correction, each involves different bias-variance tradeoffs that are well-studied in the literature, and the preferred choice will depend on the horizon length and dataset size of the specific application. We have added a brief discussion of this to the revised manuscript to help guide practitioners.
> > >
> > >
> > > **Background on conformal prediction**:
> > > In the revised manuscript, we have introduced the standard conformal prediction framework and its coverage guarantee, and described weighted conformal prediction and how importance weights correct for distribution shift.
> > >
> > > [1] Zhou, Hanhan et al. “Double Policy Estimation for Importance Sampling in Sequence Modeling Based Reinforcement Learning.”
> > >
> > > [2] Robins, J. M., Rotnitzky, A., & Zhao, L. P. (1994). Estimation of Regression Coefficients When Some Regressors are not Always Observed.
> > >
> > > [3] Angelopoulos, Anastasios N., et al. "Prediction-powered inference." (2023)

---

### Author Response · Authors · 2026-04-16
**Revised Manuscript**

We thank the reviewers for their thorough feedback. We have updated the manuscript (changes in red) to reflect this feedback.

---

### Author Response · Authors · 2026-06-08

We thank the reviewers and action editor for their feedback during this process. We have uploaded a camera ready version of the paper.

---

### Decision · Action_Editor_N7wT · 2026-05-18

**Recommendation:** Accept as is

**Audience:**

Yes

**Audience Explanation:**

Yes to quote one reviewer's summary: "The paper addresses an important and timely problem: how to perform uncertainty-aware off-policy evaluation when both real offline trajectories and synthetic/model-generated trajectories are available"

**Claims And Evidence:**

Yes

**Claims Explanation:**

Yes this is clear from all the reviews